# Bridging Vision and Language Spaces with Assignment Prediction

**Jungin Park**[1]  **Jiyoung Lee**[2*]  **Kwanghoon Sohn**[1,3*]
[1]Yonsei University   [2]NAVER AI Lab   [3]Korea Institute of Science and Technology (KIST)
{newrun, khsohn}@yonsei.ac.kr   lee.j@navercorp.com

## Abstract

This paper introduces VLAP, a novel approach that bridges pretrained vision models and large language models (LLMs) to make frozen LLMs understand the visual world. VLAP transforms the embedding space of pretrained vision models into the LLMs' word embedding space using a single linear layer for efficient and general-purpose visual and language understanding. Specifically, we harness well-established word embeddings to bridge two modality embedding spaces. The visual and text representations are simultaneously assigned to a set of word embeddings within pretrained LLMs by formulating the assigning procedure as an optimal transport problem. We predict the assignment of one modality from the representation of another modality data, enforcing consistent assignments for paired multimodal data. This allows vision and language representations to contain the same information, grounding the frozen LLMs' word embedding space in visual data. Moreover, a robust semantic taxonomy of LLMs can be preserved with visual data since the LLMs interpret and reason linguistic information from correlations between word embeddings. Experimental results show that VLAP achieves substantial improvements over the previous linear transformation-based approaches across a range of vision-language tasks, including image captioning, visual question answering, and cross-modal retrieval. We also demonstrate the learned visual representations hold a semantic taxonomy of LLMs, making visual semantic arithmetic possible.

## 1 Introduction

Vision-language models (VLMs) have achieved significant progress in demonstrating remarkable transfer and zero-shot capabilities on vision-language downstream tasks (Tan & Bansal, 2019; Lu et al., 2019; Chen et al., 2020; Huang et al., 2020; Radford et al., 2021; Jia et al., 2021). Most cutting-edge VLMs have been developed with progressively scaled-up foundation models and datasets. However, this trend underscores the demand for substantial computational resources and an extensive collection of image-text pairs. Concurrently, pretrained unimodal foundation models, such as vision transformers (Dosovitskiy et al., 2021; Caron et al., 2021; Bao et al., 2022) and large language models (LLMs) (Brown et al., 2020; Zhang et al., 2022; Raffel et al., 2020; Touvron et al., 2023; Chiang et al., 2023), have been developed with self-supervision, renewing the state-of-the-art in vision and language downstream tasks respectively. This raises the interesting question: *Can pretrained unimodal models extend their capabilities beyond the modality of pretraining data?*

Recently, intriguing attempts have been made to assemble pretrained vision models and LLMs to seem VLMs (Mokady et al., 2021; Tsimpoukelli et al., 2021; Alayrac et al., 2022; Patel & Pavlick, 2022; Eichenberg et al., 2022; Li et al., 2022; 2023; Guo et al., 2023; Liu et al., 2023a). Most of these attempts suggested ways to train relatively small-scale connection modules while freezing pretrained parameters. By doing so, it is possible to efficiently make VLMs that leverage discriminative visual representations from pretrained vision models, coupled with powerful language modeling and zero-shot capabilities of pretrained LLMs. While the burden of training costs in those methods has significantly decreased compared to pretraining VLMs from scratch, they still require high computations (both time and memory), which limits applicability (e.g. Flamingo (Alayrac et al., 2022)

---

*Corresponding authors.

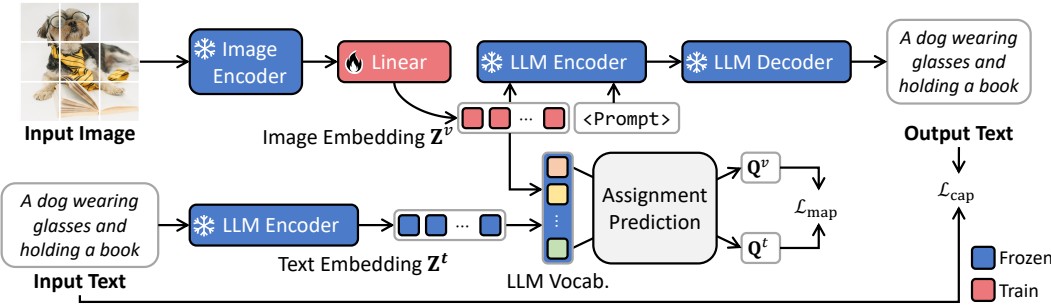

Figure 1: **Overview of VLAP**. We train a single linear layer following two learning objectives: Assignments prediction to bridge the modality gap between the visual and text representations; and image captioning to yield the generative capability of frozen LLMs.

requires 15 days with 1535 TPUs to train 10.2B parameters, BLIP-2 (Li et al., 2023) requires 9 days with 16 A100 GPUs to train 188M parameters).

Linear transformation-based approaches (Merullo et al., 2023; Koh et al., 2023) have facilitated efficient cross-modal alignment with a cost-efficient linear transformation between vision and language spaces. Merullo et al. (2023) has shown that visual representations can be linearly transformed into LLMs' input embedding as soft prompts, demonstrating that well-learned visual and text representations are functionally equivalent. Similarly, Koh et al. (2023) has proposed to generate free-form text interleaved with images by learning additional trainable tokens and linear layers. These linear transformation-based cross-modal alignments have been achieved by the image captioning objective (Merullo et al., 2023) and an additional retrieval objective (Koh et al., 2023), which directly compares the representations of image-text pairs and enforces instance discrimination as contrastive learning (Chopra et al., 2005). However, the inherent *modality gap* between different modality data leads to the inconsistency between the isolated embedding spaces (as their pretraining has been solely on unimodal data) and makes cross-modal alignment for frozen unimodal models difficult with the following challenges: (1) it relies heavily on the amount of linguistic supervision used in the pretraining of vision models, showing limited performance in image-only trained vision models (e.g. CLIP vs BEiT) (Merullo et al., 2023). (2) the contrastive objective (such as InfoNCE (van den Oord et al., 2018) in Koh et al. (2023)) is insufficient to bridge complex embedding spaces because it inherently encourages the existence of the modality gap (Liang et al., 2022).

In this paper, we propose a novel linear transformation-based VLM, which we call **VLAP**, bridging vision and language models with assignment prediction. Our VLAP learns a linear feature mapping through a novel optimal transport-based assignment prediction that maximizes the similarity between intermediate assignments for image and text data instead of directly comparing their representations. Specifically, visual and text representations are assigned to the most relevant word embeddings within pretrained LLMs. Since there is no supervision for word assignments for visual information, we cast this assignment procedure as an optimal transport problem. The assignment for one modality is predicted from the other modality representation, making two different modality representations contain the same linguistic information. We exploit the readily available word embedding space, following two main advantages: (1) it does not require additional learnable embedding space (e.g. prototypes in Asano et al. (2020b); Caron et al. (2020)) and (2) it allows visual representations to keep abundant linguistic contextual information and inherit a semantic taxonomy of LLMs. Our main contributions are four-fold:

- We present VLAP that efficiently bridges pretrained vision models and LLMs. VLAP employs an optimal transport-based assignment prediction objective to make visual and text representations compatible with each other.
- VLAP leverages the word embeddings of pretrained LLMs as a fixed central space for optimal transport. This allows us to easily bridge two pretrained frozen unimodal models by exploiting the fundamental components of LLMs.
- Mapping visual data to LLMs' word embeddings results in learned visual representations that hold a semantic taxonomy of LLMs, facilitating the operation in a textual way for visual data, e.g. visual semantic arithmetic operation.

- VLAP substantially outperforms existing linear transformation-based methods (Merullo et al., 2023; Koh et al., 2023) in a range of vision-language tasks, while also demonstrating high computational and memory efficiency.

## 2 RELATED WORK

Pretraining vision-language models (VLMs) Tan & Bansal (2019); Lu et al. (2019); Chen et al. (2020); Huang et al. (2020); Jia et al. (2021); Radford et al. (2021) has received tremendous interest thanks to their robust zero-shot capability for vision-language tasks such as image captioning, visual question answering, and cross-modal retrieval. Concurrently, in the past few years, large language models (LLMs) have achieved significant success in generating human-like language content with growth in model size and datasets at scale (Devlin et al., 2019; Radford et al., 2018; 2019; Brown et al., 2020; Raffel et al., 2020; Chung et al., 2022). To conjugate this ability of LLMs for vision-language research, recent works have attempted to transport the visual embeddings as prompts of LLMs (Tsimpoukelli et al., 2021; Eichenberg et al., 2022; Mokady et al., 2021; Alayrac et al., 2022; Sung et al., 2022; Li et al., 2023). They accomplished this by tuning the visual encoder (Tsimpoukelli et al., 2021), introducing additional lightweight modules Mokady et al. (2021); Alayrac et al. (2022); Sung et al. (2022); Cohen et al. (2022); Li et al. (2023), or employing both approaches (Eichenberg et al., 2022).

Our work is highly related to LLaVA (Liu et al., 2023a), LiMBeR (Merullo et al., 2023), and FRO-MAGe (Koh et al., 2023). Different from the modular-based methods, which require relatively a high computational cost, they learned a linear mapping to project visual embeddings into the text embedding space. They demonstrated that vision models and LLMs share non-trivial similar information even though two unimodal models are independently trained. However, their learning objectives to directly compare the visual and text representations are restrictive solutions to bridge the modality gap (Liang et al., 2022). Unlike the previous methods, we compare the intermediate assignments between image and text data by formulating an optimal transport-based assignment prediction with word embeddings of LLMs. We demonstrate that the modality gap between pretrained vision models and LLMs can be effectively bridged while preserving high computational efficiency with the proposed assignment prediction.

## 3 METHOD

Our primary goal is to make pretrained LLMs comprehend visual inputs with minimum training for general-purpose vision-language tasks. More precisely, we aim to bridge pretrained vision models and LLMs by learning a linear layer only (i.e., keeping the original parameters frozen) while preserving the representative power of pretrained vision models and the generalization ability of pretrained LLMs. To this end, we proposed a novel linear transformation-based method VLAP: bridging vision and language models with assignment prediction. The key component to bridge pretrained vision and language models is mapping information between image and text representations to make frozen LLMs interpret visual inputs as linguistic components. The previous methods (Merullo

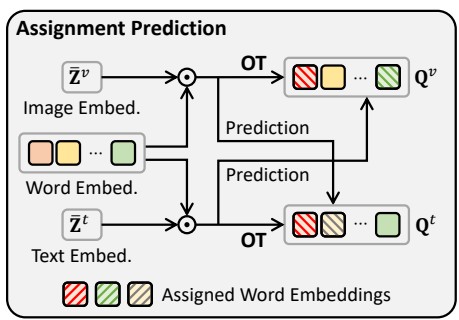

Figure 2: **Assignment prediction.** The modality gap can be relaxed by predicting the word assignments of one modality from the other modality representations.

et al., 2023; Koh et al., 2023) linearly transform the visual representation from pretrained vision model into LLMs' embedding space, and directly compare representations of paired multimodal data. In contrast, we resolve the linear cross-modal alignment using the assignment prediction objective (Asano et al., 2020b; Caron et al., 2020) that enforces consistent assignments to multimodal data. VLAP learns a linear layer by optimizing two objectives simultaneously: (1) optimal transport-based assignment prediction to enforce the visual and text representations contain the equivalent information and (2) image-to-text generation to incorporate the visual representation into the generative ability of LLMs. This section introduces each component in detail, and the overview of VLAP is shown in Figure 1.

### 3.1 Assignment Prediction

We assign visual and text representations into word embeddings by defining optimal transport as if each data is clustered into the word embedding space of pretrained LLMs. The proposed assignment prediction objective aims to predict each other's assignments. We notice that recent works for the assignment prediction problem formulate a set of learnable embeddings (i.e., prototypes) to cluster the given data (Caron et al., 2020; Xu et al., 2020; Liu et al., 2023b; Duan et al., 2022), requiring additional memory and gradient computation. Contrary to this, VLAP utilizes readily available word embeddings of LLMs as a fixed central space into which visual representations are mapped. Therefore, our training process is more stable and memory-efficient.

**Visual and text representations from pretrained models.** Given a set of $B$ images, we extract the last hidden state from pretrained image encoder as the visual representations $\mathbf{X} = \{\mathbf{X}_1, ..., \mathbf{X}_B\}$, where the representation for each image contains $d_v$-dimensional $N_p$ feature vectors, i.e., $\mathbf{X}_n = \{\mathbf{x}_n^1, ..., \mathbf{x}_n^{N_p}\} \in \mathbb{R}^{N_p \times d_v}$. Similarly, we feed a set of $B$ captions corresponding to the images into pretrained LLM to obtain the text representations $\mathbf{Z}^t = \{\mathbf{Z}_1^t, ..., \mathbf{Z}_B^t\}$, where $\mathbf{Z}_n^t = \{\mathbf{z}_n^{t,1}, ..., \mathbf{z}_n^{t,T_n}\} \in \mathbb{R}^{T_n \times d}$, $T_n$ is the number of words in the $n$-th caption, and $d$ is the embedding dimension of LLM.

**Word assignment.** We project the visual representations $\mathbf{X}$ into the same dimension as the embedding space of LLMs using a linear layer $g(\cdot)$:

$$\mathbf{Z}^v = \{\mathbf{Z}_n^v \in \mathbb{R}^{N_p \times d} \mid \mathbf{Z}_n^v = g(\mathbf{X}_n), n = 1, ..., B\}. \tag{1}$$

We assign the image and text to a set of words by mapping the averaged visual and text representations to the frozen word embeddings $\mathbf{W} = \{\mathbf{w}_1, ..., \mathbf{w}_K\}$ of LLM, where $K$ is a vocabulary size. Similar to the prior works (Asano et al., 2020b; Caron et al., 2020), we simultaneously optimize the assignment $\mathbf{Q}^v = \{\mathbf{q}_1^v, ..., \mathbf{q}_B^v\}$ and $\mathbf{Q}^t = \{\mathbf{q}_1^t, ..., \mathbf{q}_B^t\}$ to maximize the similarity between the representations for each modality and the word embeddings:

$$\max_{\mathbf{Q}^v \in \mathcal{Q}} \mathrm{Tr}(\mathbf{Q}^{v\top}\mathbf{W}^\top \bar{\mathbf{Z}}^v) + \epsilon H(\mathbf{Q}^v), \quad \max_{\mathbf{Q}^t \in \mathcal{Q}} \mathrm{Tr}(\mathbf{Q}^{t\top}\mathbf{W}^\top \bar{\mathbf{Z}}^t) + \epsilon H(\mathbf{Q}^t),$$

$$\text{where} \quad \bar{\mathbf{Z}}^v = \{\bar{\mathbf{z}}_1^v, ..., \bar{\mathbf{z}}_B^v\}, \quad \bar{\mathbf{z}}_n^v = \frac{1}{N_p}\sum_i \mathbf{z}_n^{v,i}, \quad \bar{\mathbf{Z}}^t = \{\bar{\mathbf{z}}_1^t, ..., \bar{\mathbf{z}}_B^t\}, \quad \bar{\mathbf{z}}_n^t = \frac{1}{T_b}\sum_i \mathbf{z}_n^{t,i}. \tag{2}$$

We denote the trace matrix as $\mathrm{Tr}(\cdot)$, entropy function as $H(\mathbf{Q}) = -\sum_{ij}\mathbf{Q}_{ij}\log\mathbf{Q}_{ij}$, and a smoothness parameter of the mapping as $\epsilon$. The previous methods assume that all samples are equally assigned to each cluster by constraining the matrix $\mathbf{Q}$ to belong to the transportation polytope in the whole dataset (Asano et al., 2020b) or the minibatch (Caron et al., 2020). However, the equipartition assumption can lead to impractical solutions in our work since the numbers of words are not equally distributed in practice. Therefore, we restrict the transportation polytope with the following constraints:

$$\mathcal{Q} = \{\mathbf{Q} \in \mathbb{R}_+^{K \times B} \mid \mathbf{Q}\mathbf{1}_B = \boldsymbol{\mu}_W, \mathbf{Q}^\top \mathbf{1}_K = \frac{1}{B}\mathbf{1}_B\}, \tag{3}$$

where $\mathbf{1}_B$ denotes the vector of ones in dimension $B$ and $\boldsymbol{\mu}_W$ is the marginal distribution of words, i.e., $\boldsymbol{\mu}_W(k) = \frac{N_k}{\sum_{k'} N_{k'}}$, where $N_k$ denotes the total number of the $k$-th word in the dataset. Formally, $\boldsymbol{\mu}_W$ can be represented as a vector with the length of LLM's vocabulary size (e.g., 50,272 for OPT (Zhang et al., 2022), 32,128 for T5 (Raffel et al., 2020)). By satisfying the constraints, the word embeddings are selected according to their frequency in the dataset. The optimized assignments $\mathbf{Q}^{v*}$ and $\mathbf{Q}^{t*}$ can be obtained over the set $\mathcal{Q}$ as the form of a normalized exponential matrix (Cuturi, 2013):

$$\mathbf{Q}^{v*} = \mathrm{Diag}(\boldsymbol{\mu}_W)\exp\left(\frac{\mathbf{W}^\top \bar{\mathbf{Z}}^v}{\epsilon}\right)\mathrm{Diag}(\mathbf{c}), \quad \mathbf{Q}^{t*} = \mathrm{Diag}(\boldsymbol{\mu}_W)\exp\left(\frac{\mathbf{W}^\top \bar{\mathbf{Z}}^t}{\epsilon}\right)\mathrm{Diag}(\mathbf{c}), \tag{4}$$

where $\mathbf{c}$ is a re-normalized vector in $\mathbb{R}^B$ obtained using the iterative Sinkhorn-Knopp algorithm (Cuturi, 2013).

Table 1: Performance comparisons between MAGMA (Eichenberg et al., 2022), LiMBeR (Merullo et al., 2023), and VLAP for zero-shot image captioning on the NoCaps and MSCOCO datasets. We report the architectures of visual and language models, CIDEr-D (Vedantam et al., 2015), CLIP-Score, and RefCLIP Score (Hessel et al., 2021).

| Method | Vision Encoder | LM | NoCaps (CIDEr-D) | | | | NoCaps (All) | | MSCOCO | | |
| | | | In | Out | Near | All | CLIP-S | Ref-S | CIDEr-D | CLIP-S | Ref-S |
|---|---|---|---|---|---|---|---|---|---|---|---|
| MAGMA | CLIP RN50x16 | GPT-J | 30.4 | 43.4 | 36.7 | 38.7 | 74.3 | 78.7 | 47.5 | 75.3 | 79.6 |
| LiMBeR | BEiT | GPT-J | 20.3 | 16.3 | 26.9 | 28.5 | 62.0 | 69.1 | 22.3 | 63.6 | 70.0 |
| LiMBeR | CLIP RN50x16 | GPT-J | 34.3 | 48.4 | 41.6 | 43.9 | 74.7 | 79.4 | 54.9 | 76.2 | 80.4 |
| VLAP | BEiT | $OPT_{1.3B}$ | 31.1 | 45.4 | 40.6 | 42.2 | 72.1 | 77.3 | 50.7 | 73.7 | 76.4 |
| VLAP | BEiT | $T5_{Base}$ | 31.6 | 46.3 | 41.9 | 43.4 | 72.6 | 78.9 | 51.6 | 74.2 | 78.5 |
| VLAP | CLIP ViT-B/32 | $OPT_{1.3B}$ | 48.2 | 62.7 | 59.3 | 61.3 | 84.8 | 88.5 | 69.9 | 86.7 | 91.8 |
| VLAP | CLIP ViT-B/32 | $T5_{Base}$ | 48.3 | 62.7 | 59.6 | 61.6 | 85.1 | 88.7 | 69.4 | 87.6 | 92.0 |
| VLAP | CLIP RN50x16 | GPT-J | **53.8** | **67.5** | **65.7** | **64.5** | **88.3** | **90.1** | **75.3** | **90.6** | **92.2** |

**Relaxing the modality gap with assignment prediction.** The objective of our method is to predict the assignment of one modality from the other modality representation, i.e., predicting $\mathbf{Q}^v$ from $\bar{\mathbf{Z}}^t$ and $\mathbf{Q}^t$ from $\bar{\mathbf{Z}}^v$. The assignment prediction can be formulated with the cross-entropy loss between the assignment and the probability that the corresponding modality data belongs to each word:

$$\mathcal{L}_{\text{map}} = -\frac{1}{B}\sum_{n=1}^{B}\sum_{k=1}^{K}[\mathbf{Q}_{nk}^{t}\log\mathbf{P}_{nk}^{v} + \mathbf{Q}_{nk}^{v}\log\mathbf{P}_{nk}^{t}],$$

$$\text{where} \quad \mathbf{P}_{nk}^{m} = \frac{\exp(\bar{\mathbf{z}}_{n}^{\top}\mathbf{w}_{k}/\tau)}{\sum_{k'}\exp(\bar{\mathbf{z}}_{n}^{\top}\mathbf{w}_{k'}/\tau)}, \quad m \in \{v,t\},$$

(5)

where $\tau$ is a temperature parameter (Wu et al., 2018). This loss function makes two different modality representations contain the same information.

## 3.2 IMAGE CAPTIONING WITH FROZEN LLMS

To connect the (projected) visual representations to a frozen LLM to yield the general capability of LLM (i.e., generative ability), we train the linear projection layer $g(\cdot)$ with the image captioning objective. Specifically, the visual representations $\mathbf{Z}_v$ are prepended to the input text prompt (e.g. "A photo of"), being active as soft prompts of LLM. Following the previous works (Merullo et al., 2023; Koh et al., 2023), we employ the prefix language modeling loss as the objective:

$$\mathcal{L}_{\text{cap}} = -\frac{1}{B}\sum_{n=1}^{B}\frac{1}{N_t}\sum_{t=1}^{N_t}\log f_t(s_t \mid \mathbf{Z}_n^v, [\text{prefix}], s_1, ..., s_{t-1}),$$

(6)

where $N_t$ is the number of words in the caption, $[\text{prefix}]$ is the input text prompt, and $s_t$ is a text token of the $t$-th word.

The final objective function is a weighted sum of the assignment prediction loss and captioning loss:

$$\mathcal{L} = \lambda_{\text{map}}\mathcal{L}_{\text{map}} + \lambda_{\text{cap}}\mathcal{L}_{\text{cap}},$$

(7)

where $\lambda_{\text{map}}$ and $\lambda_{\text{cap}}$ control the importance of each objective. By minimizing the final objective, we simultaneously make the visual and text representations contain the same information while keeping the capability of LLM, allowing the frozen image encoder and LLM can be effectively connected with only the linear transformation.

## 4 EXPERIMENTS

### 4.1 EXPERIMENTAL SETTINGS

We previously defined the transport polytope with the word distribution of a given dataset. This assumption poses one possible problem: VLAP often fails when the word distribution of training

Table 2: Performance comparisons between Frozen (Tsimpoukelli et al., 2021), MAGMA (Eichenberg et al., 2022), LiMBeR (Merullo et al., 2023), and VLAP for zero- and few-shot visual question answering on the VQA2 datasets. We report the architectures of visual and language models and accuracy (%).

| Method | Vision Encoder | LM | $n$-shots | | | |
|--------|---------------|-----|-----|-----|-----|-----|
| | | | 0 | 1 | 2 | 4 |
| Frozen | NFRN50 | GPT-2 | 29.5 | 35.7 | - | 38.2 |
| MAGMA | CLIP RN50x16 | GPT-J | 32.7 | 40.2 | 42.5 | 43.8 |
| LiMBeR | BEiT | GPT-J | 24.9 | 34.4 | 34.7 | 31.7 |
| LiMBeR | CLIP RN50x16 | GPT-J | 33.3 | 39.9 | 40.8 | 40.3 |
| VLAP | BEiT | $OPT_{1.3B}$ | 34.5 | 44.2 | 45.1 | 45.7 |
| VLAP | BEiT | $T5_{Base}$ | 34.7 | 43.3 | 45.4 | 45.5 |
| VLAP | CLIP ViT-B/32 | $OPT_{1.3B}$ | 40.4 | **52.6** | **53.8** | **54.7** |
| VLAP | CLIP ViT-B/32 | $T5_{Base}$ | **41.1** | 51.3 | 51.9 | 52.6 |

Table 3: Performance comparisons between ViLBERT (Lu et al., 2019), CLIP (Radford et al., 2021), ESPER (Yu et al., 2022), FROMAGe (Koh et al., 2023), and VLAP for zero-shot image-and-text-to-text (IT2T) and text-to-image (T2I) retrieval on the Visual Dialog datasets. Following Koh et al. (2023), we report Normalized Discounted Cumulative Gain (NDCG), Mean Reciprocal Recall (MRR), R@1, R@5, and R@10 for IT2T retrieval and R@1, R@5, and R@10 for T2I retrieval.

| Method | Train. Params. | Training Data | IT2T | | | | | T2I | | |
|--------|---------------|--------------|------|-----|-----|-----|------|-----|-----|------|
| | | | NDCG | MRR | R@1 | R@5 | R@10 | R@1 | R@5 | R@10 |
| ViLBERT | 114M | 3.1M | 11.6 | 6.9 | 2.6 | 7.2 | 11.3 | - | - | - |
| CLIP ViT-L/14 | 300M | 400M | 10.9 | 8.5 | 3.1 | 8.7 | 15.9 | 17.7 | 38.9 | 50.2 |
| ESPER | 4M | 0.5M | 22.3 | **25.7** | 14.6 | - | - | Incapable | | |
| FROMAGe | 5.5M | 3.1M | 16.5 | 22.0 | 17.6 | 20.1 | 25.1 | 20.8 | 44.9 | 56.0 |
| VLAP w/CLIP-$T5_{Base}$ | 0.6M | 3.1M | 20.5 | 24.9 | **21.1** | **23.7** | **29.6** | **26.9** | **55.3** | **69.1** |

data cannot cover real-world scenarios. To prevent this issue, we preserve the soft assignments as in Caron et al. (2020) instead of strictly assigning representations to words included in the given caption data. In addition, we carefully argue that a large dataset (e.g. CC3M (Sharma et al., 2018)) is sufficient to cover the real-world scenario. To validate this, we mainly investigate vision-language tasks with the zero-shot setting, i.e., training and test data come from different datasets.

**Datasets.** We evaluate VLAP on zero-shot image captioning, visual question answering (VQA), and cross-modal retrieval. We first train the model on the CC3M (Sharma et al., 2018) and evaluate the performance on the following datasets for each task. For zero-shot image captioning, we evaluate the performance on MSCOCO (Lin et al., 2014) and NoCaps (Agrawal et al., 2019), following (Merullo et al., 2023). For visual question answering, we evaluate the model on the VQA2 (Goyal et al., 2017) dataset from zero-shot to 4-shot settings. In cross-modal retrieval, we use the Visual Dialog (Das et al., 2017) dataset for the comparability to previous work (Koh et al., 2023). The input text prompt for each task is presented in Section B.

**Model architecture.** For the frozen image encoders, we employ two vision transformer models according to their pretraining data configuration: BEiT (Bao et al., 2022) pretrained on image-only data and CLIP ViT/B-32 (Radford et al., 2021) jointly learned with the text encoder on image-text data. For the frozen language models, we explore two types of LLMs according to their architectural configuration: $OPT_{1.3B}$ (Zhang et al., 2022) as decoder-based LLMs and $T5_{Base}$ (Raffel et al., 2020) as encoder-decoder-based LLMs.

## 4.2 ZERO-SHOT IMAGE CAPTIONING

We provide performance comparisons between Frozen (Tsimpoukelli et al., 2021), MAGMA (Eichenberg et al., 2022), LiMBeR (Merullo et al., 2023), and VLAP for zero-shot image captioning in Table 1. We mainly report CIDEr-D (Vedantam et al., 2015), CLIPScore, and Ref-CLIPScore (Hessel et al., 2021), following (Merullo et al., 2023). The results show that

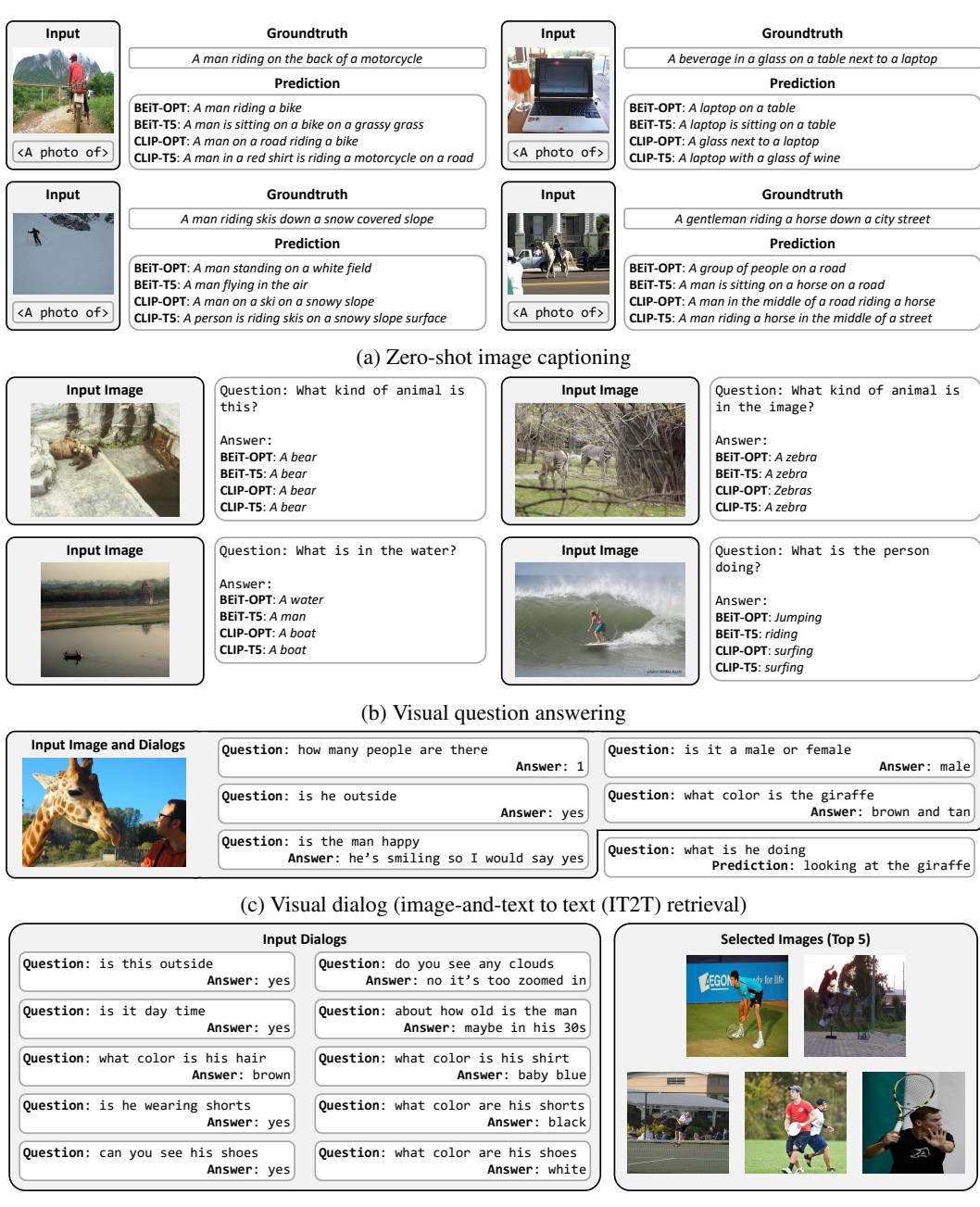

Figure 3: Selected examples from VLAP for vision-language tasks, including (a) zero-shot image captioning, (b) visual question answering (VQA), (c) visual dialog, and (d) text-to-image (T2I) retrieval.

VLAP consistently outperforms the previous linear transformation-based approach (Merullo et al., 2023) with large margins. Especially, VLAP with the BEiT with T5$_{Base}$ achieves 14.9% and 29.3% CIDEr-D improvements on each dataset. It achieves comparable performance to LiMBeR with the CLIP image encoder, despite BEiT lacking linguistic supervision in its pretraining. VLAP also attains significantly higher CLIPScores, which represent how the image and generated caption are semantically similar, improving the CLIP-S by 10.4%, 11.4%, and the RefCLIP-S by 9.3%, 11.6% on NoCaps and MSCOCO, respectively. In addition, we use a much smaller number of parameters in LLMs (T5$_{Base}$ has 220M parameters, while GPT-J has 6B parameters), demonstrating the effectiveness of the proposed method in grounding LLMs to visual data. In practice, VLAP with the CNN-based vision model (i.e., CLIP RN50x16 (Radford et al., 2021)) and the larger LLM (i.e., GPT-J (Wang & Komatsuzaki, 2021)) significantly outperforms MAGMA and LiMBeR with

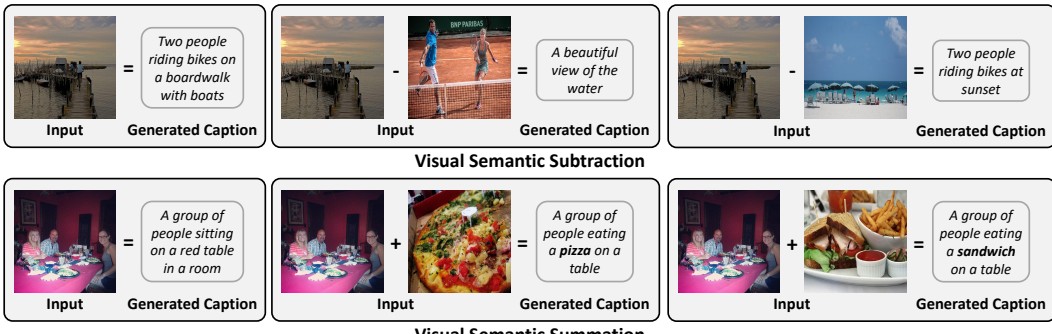

Figure 4: Selected examples for visual semantic arithmetic.

the same backbones by large margins across all metrics. The performance evaluated with additional captioning metrics, including BLEU (Papineni et al., 2002), METEOR (Banerjee & Lavie, 2005), ROUGE (Lin, 2004), and SPICE (Anderson et al., 2016), are shown in Appendix C.

Qualitative examples are illustrated in Figure 3a. The results show impressive performance of VLAP across all vision-language models. VLAP with the BEiT consistently generates apposite captions, even if occasionally produces relatively simplified captions and provides some wrong words (e.g. 'snow' as 'white field' or 'air' in the example).

### 4.3 VISUAL QUESTION ANSWERING

We evaluate the performance with zero-shot and few-shot settings[1] on visual question answering (VQA). In $n$-shot settings, we randomly prepend $n$ complete examples before each question as described in Tsimpoukelli et al. (2021); Eichenberg et al. (2022). We provide the comparison between VLAP and three baselines (Tsimpoukelli et al., 2021; Eichenberg et al., 2022; Merullo et al., 2023), as shown in Table 2. VLAP outperforms the previous methods with large margins, showing the effectiveness of the proposed method. Surprisingly, VLAP with the BEiT and $T5_{Base}$, which is the most challenging condition (in terms of pretraining data and the number of parameters), achieves higher performance than the previous methods for all evaluation settings.

In Figure 3b, we provide the qualitative examples for zero-shot and 4-shot VQA. For zero-shot VQA, the results show that while VLAP generally infers the correct answers with all image encoders and language models, some failure cases are shown with the BEiT models.

### 4.4 CROSS-MODAL RETRIEVAL

For cross-modal retrieval, we compare the performance of ViLBERT (Lu et al., 2019), CLIP (Radford et al., 2021), ESPER (Yu et al., 2022), FROGMAGe (Koh et al., 2023), and VLAP evaluated on Visual Dialog (Das et al., 2017) for two types of retrieval tasks: (1) visual dialog referred to as image-and-text-to-text (IT2T) retrieval to reason the correct answer from given an image, a question, and a dialog about an image. We report the Normalized Discounted Cumulative Gain (NDCG), Mean Reciprocal Recall (MRR), and Recall@k, following Yu et al. (2022); Koh et al. (2023). (2) text-to-image (T2I) retrieval to find the image from a given dialog. We report the Recall@k performance for T2I retrieval. Since a dialog in the dataset consists of question-answer pairs for images, we use the text prompt used in the VQA task. We provide the comparisons for cross-modal retrieval in Table 3.

**Image-and-text to text (IT2T) retrieval.** For IT2T retrieval, we compute the loss (i.e. cross-entropy loss) between the prediction and given answer candidates, and select the answer with the lowest loss. VLAP outperforms the previous methods (Lu et al., 2019; Radford et al., 2021; Yu et al., 2022; Koh et al., 2023), achieving 3.5%, 3.6%, and 4.5% improvements on R@1, 5, and 10, respec-

---

[1]We follow the procedure of the official VQA 2.0 repo: `https://github.com/GT-Vision-Lab/VQA`.

tively. While VLAP shows slightly lower NDCG and MRR performance than ESPER (Yu et al., 2022), they train the model on the superset of Visual Dialog (i.e. MSCOCO), already containing in-context information of the dataset. The comparison between VLAP and FROMAGe demonstrates the effectiveness of our assignment prediction objective as the main difference between the two methods is the learning objective. VLAP consistently outperforms FROMAGe (Koh et al., 2023) even with fewer trainable parameters, improving 4.0% and 2.9% on NDGC and MRR, respectively.

**Text-to-image (T2I) retrieval.** As VLAP minimizes the modality gap between image and text data, the visual and text representations can be directly used to retrieve each other. For T2I retrieval, we first extract the text and visual representations for the given dialog and all images in the dataset. We directly measure the similarity between the text and visual representations and select the image with the highest similarity. VLAP substantially outperforms the prior works (Radford et al., 2021; Koh et al., 2023) over all evaluation metrics, achieving 9.2% and 6.1% improvements on R@1 over CLIP (Radford et al., 2021) and FROMAGe.

The experiments on cross-modal retrieval demonstrate the flexibility of VLAP for vision-language tasks. We emphasize that VLAP provides competitive results even on the retrieval task, showing that VLAP can be applied as a general model without being limited to tasks.

## 4.5 VISUAL SEMANTIC ARITHMETIC

The pretrained LLMs capture task-agnostic linguistic structures and lexical semantics (Jawahar et al., 2019; Liu et al., 2019; Tenney et al., 2019; Hewitt & Manning, 2019; Vulić et al., 2019). Recent work (Tewel et al., 2022) has shown this capability persists in VLMs so that the visual embeddings can be expressed in a textual way, which is called visual semantic arithmetic. VLAP could perform this task because the visual representations are learned to hold a semantic taxonomy of LLMs. For example, the semantic direction between two images can be expressed by subtracting one visual representation from another representation. Similarly, the summation of two visual representations allows for guidance for text generation. To demonstrate the arithmetical ability of VLAP, we provide an analysis of visual semantic arithmetic (i.e., subtraction and summation operations) in Figure 4. In subtraction, the conceptual direction between two images can be obtained. For example, "A beautiful view of the water" and "Two people riding bikes at sunset" are obtained from the same image of "Two people riding bikes on a boardwalk with boats" by subtracting images containing "two people" and "view of the ocean," respectively. Meanwhile, the visual concepts can be guided by other visual semantics with summation operations. For example, a scene of people sitting around a table can be guided by the image of "pizza" and "sandwich", generating a caption of "A group of people eating a pizza (or sandwich) on a table". These results demonstrate that the visual representations from VLAP contain the semantic taxonomy of LLMs, allowing the VLAP's visual representations to be expressed in a textual way.

## 5 CONCLUSION AND FUTURE WORK

We propose VLAP to bridge pretrained vision encoders and LLMs using a single linear layer for vision-language tasks. VLAP efficiently and effectively learns the linear mapping between image and text data by formulating assignment prediction using LLMs' word embeddings as the optimal transport. Experimental findings corroborate the effectiveness of VLAP, demonstrating significant performance compared to the previous linear transformation-based methods on a spectrum of vision-language tasks. Furthermore, we demonstrate that VLAP holds a semantic taxonomy of LLMs, highlighting the emerging flexibility and applicability of the proposed approach.

While linear transformation-based methods excel in computation and memory efficiency, there is still a substantial performance gap against modular-based methods, such as Flamingo and BLIP-2. We attribute such gap to a lot of trainable parameters in their methods (10.2B in Flamingo, 188M in BLIP-2) and larger training data (1.8B image-text pair in Flamingo, 129M image-text pair in BLIP-2). Since the optimal transport-based assignment prediction can be easily moved to the modular-based methods, scaling VLAP with the modular-based models and training on larger multimodal datasets are promising directions for future work.

## 6 REPRODUCIBILITY STATEMENT

VLAP employs the pretrained BEiT and CLIP image encoder as vision models and the pretrained OPT$_{1.3B}$ and T5$_{Base}$ as LLMs, which can be accessed by anyone. We provide PyTorch implementation for VLAP at `https://github.com/park-jungin/vlap`. The implementations will enable researchers to reproduce the results demonstrated in the paper as well as conduct additional analysis for VLAP with other vision models and LLMs.

## ACKNOWLEDGEMENT

This research was supported by the National Research Foundation of Korea (NRF) grant funded by the Korea government (MSIP) (NRF2021R1A2C2006703).

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

Table 4: Hyperparameters for training VLAP corresponding to each image encoders and LLMs.

| Hyperparameter | Model | | | |
|---|---|---|---|---|
| | BEiT-OPT$_{1.3B}$ | BEiT-T5$_{Base}$ | CLIP-OPT$_{1.3B}$ | CLIP-T5$_{Base}$ |
| Warmup steps | 1.5K | 3K | 1.5K | 3K |
| Learning rate | 1e-4 | 5e-3 | 1e-4 | 5e-3 |
| Batch size | 128 | 256 | 128 | 256 |
| Total steps | 30K | 15K | 30K | 15K |
| Final learning rate | | 0 | | |
| AdamW $\beta$ | | (0.9, 0.999) | | |
| Text prompt | | `A photo of` | | |

## A  TRAINING DETAILS

In Table 4, we provide hyperparameters used in training.

## B  INFERENCE DETAILS

As demonstrated in Section 4, VLAP presents the emerging flexibility and applicability to various vision-language tasks. While VLAP is trained on a unified training procedure, the inference provides output with a slightly different scheme with respect to the task. In Figure 5, we provide illustrations for inference corresponding to each task to prevent confusion and help clearly understand the inference procedure of VLAP.

### B.1  IMAGE CAPTIONING

For image captioning, VLAP provides the text description for the given image. As shown in Figure 5a, we use "`A photo of`" as the input text prompt.

### B.2  VISUAL QUESTION ANSWERING

For VQA, we use the text prompt "`Question: {} Answer:`" for the OPT model and "`Question: {} Short Answer:`" for the T5 model, following Li et al. (2023).

### B.3  CROSS-MODAL RETRIEVAL

**Visual dialog (IT2T retrieval).**  In visual dialog (IT2T retrieval), the image and the sequence of question-answer dialog are given as inputs. We prepend the visual representations to text representations of the dialog sequence and feed them into LLMs. As shown in Figure 5c, we measure the cross-entropy loss between the generated output and the answer candidates, and select the one with the lowest loss as the prediction.

**Text-to-image (T2I) retrieval.**  Different from the other tasks, VLAP does not leverage the generative ability of LLMs for T2I retrieval. As shown in Figure 5d, we extract the visual representations for all images in the dataset. We also extract the text representations for the input dialog and directly measure the similarity between the text representations and all visual representations. We select the target image with the highest similarity.

## C  ADDITIONAL RESULTS

### C.1  IMAGE CAPTIONING

We provide the zero-shot image captioning performance on the NoCaps (Agrawal et al., 2019) and MSCOCO (Lin et al., 2014) datasets with additional evaluation metrics, including BLEU (Papineni et al., 2002), METEOR (Banerjee & Lavie, 2005), ROUGE (Lin, 2004), and SPICE (Anderson

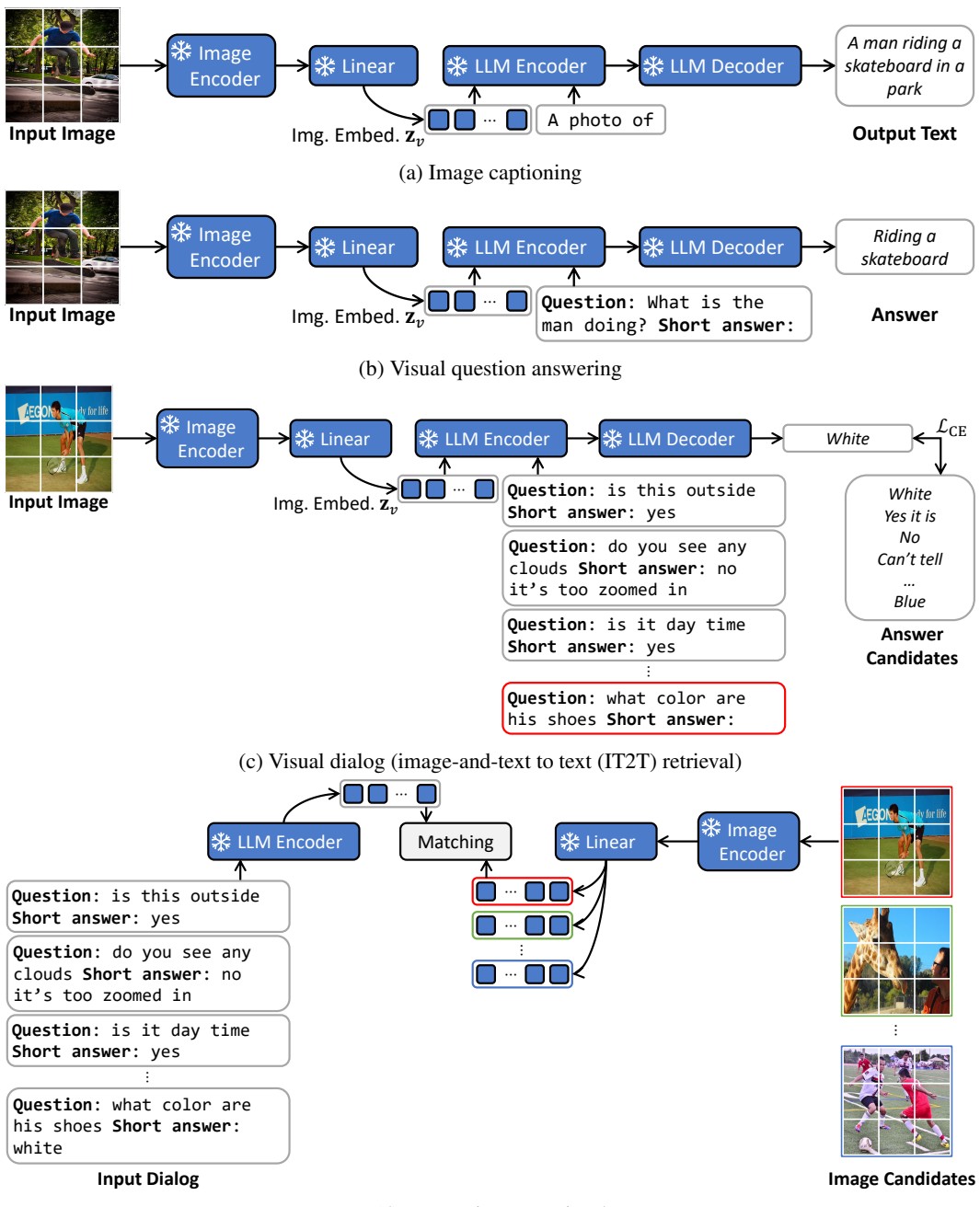

Figure 5: Illustrations for the inference on (a) image captioning, (b) visual question answering, (c) visual dialog, and (d) text-to-image retrieval.

et al., 2016). As shown in Table 5, VLAP with CLIP ViT-B/32 (Radford et al., 2021) and two LLMs (Zhang et al., 2022; Raffel et al., 2020) persistently outperform the previous methods, including MAGMA (Eichenberg et al., 2022), LiMBeR (Merullo et al., 2023), and FROMAGe (Koh et al., 2023). In addition, VLAP with BEiT (Bao et al., 2022) achieves comparable or better performance on several evaluation metrics than the previous approaches with CLIP image encoders, demonstrating the effectiveness of the proposed assignment prediction objective.

Table 5: Performance comparisons between MAGMA (Eichenberg et al., 2022), LiMBeR (Merullo et al., 2023), FROMAGe (Koh et al., 2023), and VLAP for zero-shot image captioning on the MSCOCO dataset. We report the architectures of visual and language models, BLEU (Papineni et al., 2002), METEOR (Banerjee & Lavie, 2005), ROUGE (Lin, 2004), and SPICE (Anderson et al., 2016).

| Method | Vision Encoder | LM | BLEU-1 | BLEU-2 | BLEU-3 | BLEU-4 | METEOR | ROUGE | SPICE |
|---|---|---|---|---|---|---|---|---|---|
| MAGMA | CLIP RN50x16 | GPT-J | 0.432 | 0.300 | 0.203 | 0.137 | 0.159 | 0.376 | 0.117 |
| LiMBeR | BEiT | GPT-J | 0.319 | 0.187 | 0.105 | 0.060 | 0.106 | 0.299 | 0.093 |
| LiMBeR | CLIP RN50x16 | GPT-J | 0.400 | 0.278 | 0.187 | 0.126 | 0.161 | 0.376 | 0.121 |
| FROMAGe | CLIP ViT-L/14 | $OPT_{6.7B}$ | 0.477 | 0.293 | 0.172 | 0.102 | 0.287 | - | - |
| VLAP | BEiT | $OPT_{1.3B}$ | 0.449 | 0.348 | 0.225 | 0.174 | 0.280 | 0.369 | 0.118 |
| VLAP | BEiT | $T5_{Base}$ | 0.458 | 0.357 | 0.266 | 0.176 | 0.287 | 0.382 | 0.120 |
| VLAP | CLIP ViT-B/32 | $OPT_{1.3B}$ | 0.567 | 0.429 | 0.343 | 0.291 | 0.310 | 0.482 | 0.135 |
| VLAP | CLIP ViT-B/32 | $T5_{Base}$ | **0.571** | **0.437** | **0.348** | **0.297** | **0.311** | **0.487** | **0.137** |

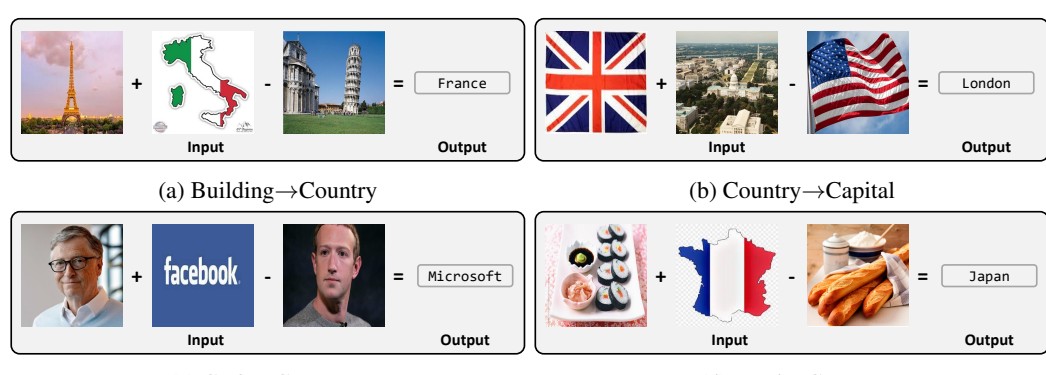

(a) Building→Country      (b) Country→Capital

(c) CEO→Company      (d) Food→Country

Figure 6: Illustrations for the visual relation benchmark.

## C.2 VISUAL SEMANTIC ARITHMETIC

**Performance on the visual relation benchmark.** To further explore the capability in visual semantic arithmetic of VLAP, we evaluate VLAP on the visual relation (VR) benchmark (Tewel et al., 2022). The VR benchmark consists of a total of 320 relations, including building→country, country→capital, CEO→company, food→country, and leader→country. Since the relation of leader→country includes out-of-date information (e.g. 'Obama'-'USA'), we exclude it from this experiment. We illustrate examples for the VR benchmark in Figure 6. In Table 6, we compare the performance between VLAP, ClipCap (Mokady et al., 2021), and ZeroCap (Tewel et al., 2022) in terms of BLEU-1 (Papineni et al., 2002), Recall@5, and CLIP-Score (Hessel et al., 2021), following Tewel et al. (2022). For fair comparisons, we identically use CLIP ViT-B/32 (Radford et al., 2021) as the image encoder and GPT-2 (Radford et al., 2019) as the LLM. The results show that VLAP outperforms the previous methods by large margins. In particular, we attain high correlations of over 75% in all relations on the semantic distance-based metric, i.e., CLIP-Score.

**Visualization.** We provide UMAP visualization (Sainburg et al., 2021) of visual representations to analyze the visual arithmetic analogy of VLAP in feature space, as shown in Figure 7. The blue and green circles are the real pairs and the red circles denote the embeddings obtained by the visual arithmetic operation. For example, the images of 'the Eiffel Tower'-'France' and 'the Leaning Tower of Pisa'-'Italy' in Figure 7a have a 'building-country' relation. The direction from 'building' to 'country' can be obtained through a subtraction operation between visual features of 'Italy' and 'the Leaning Tower of Pisa.' The summation of this direction with the embedding of 'the Eiffel Tower' allows us to derive the embedding corresponding to the same relation as 'building-country,' i.e., 'France.' In practice, the embedding obtained by visual arithmetic operations is close to the image of 'France' in the feature space and the LLM generates the caption of 'A photo of France' with the embedding obtained by visual arithmetic operations. Similarly, the direction for

Table 6: Performance comparisons between ClipCap (Mokady et al., 2021), ZeroCap (Tewel et al., 2022), and VLAP for visual semantic arithmetic on the Visual Relation dataset. Following Tewel et al. (2022), we report BLEU-1, R@5, and CLIP-Score.

| Method | Building $\rightarrow$ Country | | | Country $\rightarrow$ Capital | | | CEO $\rightarrow$ Company | | | Food $\rightarrow$ Country | | |
|--------|------|------|--------|------|------|--------|------|------|--------|------|------|--------|
| | B@1 | R@5 | CLIP-S | B@1 | R@5 | CLIP-S | B@1 | R@5 | CLIP-S | B@1 | R@5 | CLIP-S |
| ClipCap | 0.003 | 0.035 | 0.24 | 0.0 | 0.0 | 0.22 | 0.004 | 0.005 | 0.18 | 0.0 | 0.0 | 0.24 |
| ZeroCap | 0.1 | 0.32 | 0.7 | 0.14 | 0.32 | 0.68 | 0.1 | 0.3 | 0.64 | 0.03 | 0.33 | 0.66 |
| VLAP | **0.3** | **0.53** | **0.87** | **0.46** | **0.62** | **0.89** | **0.25** | **0.5** | **0.75** | **0.17** | **0.5** | **0.81** |

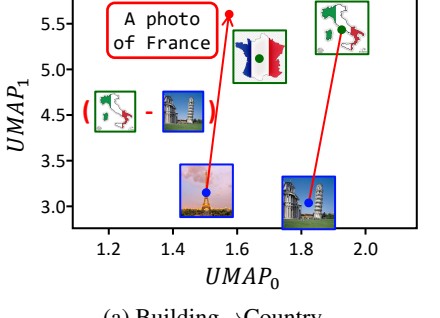
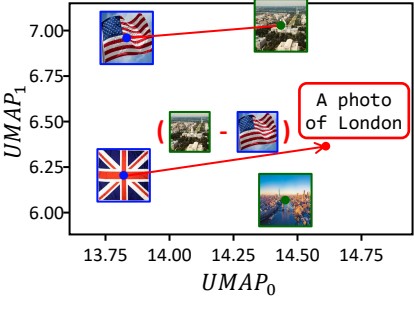

(a) Building→Country                     (b) Country→Capital

Figure 7: UMAP visualization (Sainburg et al., 2021) of selected examples from the visual relation benchmark.

the relation of 'country-capital' can be obtained by subtracting the image of 'USA' from the image of 'Washington.' The LLM generates the caption 'A photo of London' from the embedding of the summation of the direction and the image of 'England.'

# D ABALATION STUDY

## D.1 LOSS FUNCTION

Table 7: Zero-shot image captioning performance on MSCOCO (Lin et al., 2014) corresponding to (a) different ratios between the balancing parameters (i.e., $\lambda_{map} : \lambda_{cap}$) and (b) combinations of learning objectives.

| $\lambda_{map}$ | $\lambda_{cap}$ | CIDEr-D | CLIP-S | Ref-S |
|------|------|---------|--------|-------|
| 0 | 1 | 51.7 | 72.1 | 76.3 |
| 0.2 | 0.8 | 62.3 | 80.3 | 85.0 |
| 0.4 | 0.6 | 68.1 | 84.8 | 89.2 |
| 0.5 | 0.5 | 69.6 | 86.3 | 91.6 |
| 0.6 | 0.4 | 69.9 | 86.7 | 91.8 |
| 0.8 | 0.2 | 64.5 | 83.1 | 87.8 |
| 1 | 0 | 49.6 | 72.4 | 76.1 |

(a) Different ratios between $\lambda_{map}$ and $\lambda_{map}$

| $\mathcal{L}_{cap}$ | $\mathcal{L}_{ITM}$ | $\mathcal{L}_{ITC}$ | $\mathcal{L}_{map}$ | CIDEr-D | CLIP-S | Ref-S |
|------|------|------|------|---------|--------|-------|
| ✓ | | | | 51.7 | 72.1 | 76.3 |
| ✓ | ✓ | | | 58.4 | 78.2 | 81.7 |
| ✓ | ✓ | ✓ | | 61.8 | 79.0 | 82.3 |
| ✓ | ✓ | ✓ | ✓ | 65.7 | 82.5 | 85.8 |
| ✓ | | | ✓ | 69.9 | 86.7 | 91.8 |

(b) Combinations of training objectives

To validate the effectiveness of each component in VLAP, we evaluate the zero-shot image captioning performance of VLAP trained with various combinations of learning objectives on MSCOCO (Lin et al., 2014). All experiments are conducted with CLIP ViT-B/32 (Radford et al., 2021) and OPT$_{1.3B}$ (Zhang et al., 2022) as the vision model and LLM, respectively.

**Effectiveness of $\lambda_{map}$ and $\lambda_{cap}$.** The balancing parameters in equation 7 control the importance of $\mathcal{L}_{map}$ and $\mathcal{L}_{cap}$. In Table 7a, we report the performance corresponding to different ratios between $\lambda_{map}$ and $\lambda_{cap}$ while keeping the sum of two terms as 1. In the first row (i.e., $\lambda_{map} : \lambda_{cap} = 0 : 1$), the final objective is $\mathcal{L} = \mathcal{L}_{cap}$, which is equivalent to the language modeling objective of LiM-BeR (Merullo et al., 2023) and the first stage of LLaVA (Liu et al., 2023a). With this objective only,

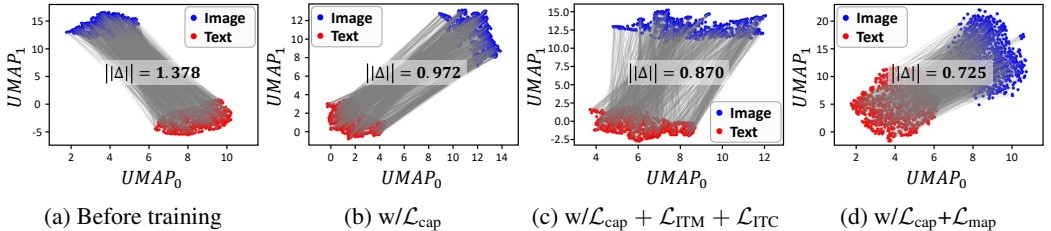

Figure 8: UMAP visualization (Sainburg et al., 2021) of visual and text representations corresponding to the combinations of objectives.

VLAP performs significantly poorly, attaining 51.7 of CIDEr-D. While VLAP with only assignment prediction (i.e., $\mathcal{L} = \mathcal{L}_{\text{map}}$) performs worse than with only language modeling, $\mathcal{L}_{\text{map}}$ improves the performance even by a small portion, showing a performance improvement of 10.6 on the CIDEr-D score. Increasing the ratio of $\lambda_{\text{map}}$ gradually improves the performance, achieving the best performance of 69.9 with $\lambda_{\text{map}} : \lambda_{\text{cap}} = 0.6 : 0.4$. Thus, the results demonstrate the effectiveness of the proposed assignment prediction objective while emphasizing the necessity of the language modeling objective to leverage the generative ability of LLMs.

**With additional objectives.** Image-text contrastive learning (ITC) and image-text matching (ITM) objectives, which directly compare image and text representations, have been widely used in cross-modal alignment (Li et al., 2021; 2022; 2023). We use the ITC and ITM objectives to evaluate the performance of VLAP trained with these objectives. For ITC, we contrast the similarity between averaged image and text representations (i.e., $\bar{\mathbf{Z}}^v$ and $\bar{\mathbf{Z}}^t$) of a positive pair against those of negative pairs in a batch. The ITC objective $\mathcal{L}_{\text{ITC}}$ is defined as the cross-entropy loss between pairwise similarities and the groundtruth one-hot similarities, where the positive pair has 1 and the negative pairs have 0. For ITM, we learn an additional linear layer as a binary classifier to determine whether an image-text pair is matched or not. We concatenate the averaged image and text representations and feed them into the classifier with the softmax activation. Following Li et al. (2021; 2022; 2023), we also employ the hard negative sampling strategy based on the pairwise similarity. The ITM objective $\mathcal{L}_{\text{ITM}}$ is defined as the binary cross-entropy loss.

We evaluate VLAP trained with different combinations of learning objectives: (1) $\mathcal{L}_{\text{cap}}$; (2) $\mathcal{L}_{\text{cap}} + \mathcal{L}_{\text{ITM}}$; (3) $\mathcal{L}_{\text{cap}} + \mathcal{L}_{\text{ITM}} + \mathcal{L}_{\text{ITC}}$; (4) $\mathcal{L}_{\text{cap}} + \mathcal{L}_{\text{ITM}} + \mathcal{L}_{\text{ITC}} + \mathcal{L}_{\text{map}}$; and (5) the original objective of VLAP, $\mathcal{L}_{\text{cap}} + \mathcal{L}_{\text{map}}$. As shown in Table 7b, $\mathcal{L}_{\text{ITM}}$ and $\mathcal{L}_{\text{ITC}}$ contribute to performance improvement with $\mathcal{L}_{\text{cap}}$, achieving 58.4 in (2) and 61.8 in (3). However, we notice that the ITM and ITC objectives hurt overall performance, as shown in the performance comparison between (4) and (5). We argue that ITM and ITC solely focus on direct pairwise comparisons without considering the semantic correlation between predefined two embedding spaces. Namely, they overly enforce instance discrimination to encourage the existence of the modality gap (Liang et al., 2022).

To further analyze the impact of each objective, we also provide UMAP visualization (Sainburg et al., 2021) with the modality gap distance, as shown in Figure 8. Note that the blue and red circles represent image and text embeddings and the gray lines refer to image-text pairs, respectively. Formally, the modality gap $\Delta$ is defined as the difference between the center of visual and text representations (Liang et al., 2022):

$$\Delta = \frac{1}{n} \sum_{i=1}^{n} \bar{\mathbf{z}}_i^v - \frac{1}{n} \sum_{i=1}^{n} \bar{\mathbf{z}}_i^t, \tag{8}$$

where $n$ is the total number of image-text pairs, $\bar{\mathbf{z}}_i^v$, and $\bar{\mathbf{z}}_i^t$ are the $i$-th averaged visual and text representations. As shown in Figure 8a, we can observe the obvious modality gap before training, which separates two modality embeddings, and the default gap distance between CLIP ViT-B/32 and T5$_{\text{Base}}$ is $||\Delta|| = 1.378$. Although the image captioning objective $\mathcal{L}_{\text{cap}}$ (Figure 8b) and the additional objectives (Figure 8c) reduce the modality gap distance to $||\Delta|| = 0.972$ and $||\Delta|| = 0.870$, respectively, the modality gap still exists. In Figure 8d, the modality gap between two modality embeddings is significantly relaxed and the gap distance is further reduced to $||\Delta|| = 0.725$ with our assignment prediction objective.

Table 8: Ablation study for the components in assignment prediction. We evaluate the zero-shot image captioning performance on MSCOCO (Lin et al., 2014) and report the CIDEr-D score.

| | Method | $\epsilon-$MSCOCO (CIDEr-D) | | | |
|---|---|---|---|---|---|
| | | 0.1 | 0.05 | 0.01 | 0.005 |
| (i) | VLAP w/o word embeddings | 20.4 | 43.8 | 38.9 | - |
| (ii) | VLAP w/o word distribution | 50.7 | 57.9 | 57.1 | 46.2 |
| (iii) | VLAP | 61.8 | 68.0 | 69.9 | 67.7 |

Table 9: Performance comparisons of VLAP according to the training dataset for several zero-shot vision-language tasks.

| Training Data | Vision Encoder | LM | NoCaps (CIDEr-D) | MSCOCO (CIDEr-D) | VQA2 ($n = 0$) | IT2T (R@1) | T2I (R@1) |
|---|---|---|---|---|---|---|---|
| CC3M | CLIP ViT-B/32 | T5$_{Base}$ | 61.6 | 69.4 | 41.1 | 21.1 | 26.9 |
| CC12M | CLIP ViT-B/32 | T5$_{Base}$ | 62.4 | 70.1 | 41.3 | 23.0 | 28.2 |

## D.2 OPTIMAL TRANSPORT

The assignment prediction of VLAP mainly differs from the previous optimal transport-based (or clustering-based) approaches (Asano et al., 2020b; Caron et al., 2020; Asano et al., 2020a; Duan et al., 2022) in the following aspects: (1) defining a fixed central space instead of a learnable central space and (2) defining the transportation polytope with the word distribution of the training dataset instead of the equipartition assumption. We conduct additional experiments to validate the effectiveness of each component in our assignment prediction. In Table 8, we evaluate each model for zero-shot image captioning (CIDEr-D) on MSCOCO corresponding to the varying entropy parameter $\epsilon$. In (i) "VLAP w/o word embeddings", we train VLAP with learnable prototypes as in Asano et al. (2020b); Caron et al. (2020); Asano et al. (2020a); Duan et al. (2022). In this setting, we define 3K prototypes following Caron et al. (2020) and apply the equipartition assumption. In (ii) "VLAP w/o word distribution", we also verify the effectiveness of the word distribution. In this setting, we use the word embeddings as a fixed central space and perform the optimal transport with the equipartition assumption. The comparison between (i) and (ii) shows that the word embedding achieves substantial performance improvement, demonstrating the effectiveness of the word embedding. The comparison between (ii) and (iii) original VLAP shows that the transportation polytope with the word distribution achieves additional performance improvement and provides more robust performance regarding $\epsilon$.

## D.3 TRAINING DATASETS

To verify the impact of the training dataset scale on the zero-shot capability, we train VLAP on the CC12M (Changpinyo et al., 2021) dataset, which contains about $4\times$ more images than CC3M. We report the performance on zero-shot vision-language tasks in Table 9. The performance of VLAP is marginally improved across all tasks despite the considerable increase in the amount of training data. Since VLAP defines transportation polytope with a marginal word distribution, simply increasing the number of training data while keeping a similar word distribution could not derive a significant performance improvement. Namely, VLAP has training efficiency for a relatively small amount of data that might cover a word distribution.

