# OpenReview forum: "Bridging Vision and Language Spaces with Assignment Prediction"
_ICLR.cc/2024/Conference — ICLR 2024 poster_

### Official Review · Reviewer_zAaX · 2023-10-28

**Soundness:** 2 fair
**Presentation:** 3 good
**Contribution:** 2 fair
**Rating:** 5
**Confidence:** 4

**Summary:**

The paper proposes a method to align the visual representations of pretrained visual encoders into the input space of pretrained language models, using a linear projection layer. The linear layer is the only trainable part in the system, which is supervised by two losses: (a) assignment consistency - the visual features and text features are assigned to the word, and a similarity loss between the assignment results is applied (b) an image captioning objective. Using this method, experiments are done on 3 tasks, including image captioning, VQA, image-text retrieval to show that the method outperforms existing methods. Different variations of visual and text models are studied in the experiments. Additionally, some qualitative visual semantic arithmetic results are provided.

**Strengths:**

1. The method is simple and clear - train a linear layer with two losses including the newly proposed assignment prediction loss.
2. Intensive experiments are provided on 3 tasks using different visual backbones (CLIP, BeiT) and text backbones (OPT-1.3B and T5-base), where results consistently outperforms existing methods.
3. The paper is well-written and easy to follow.

**Weaknesses:**

My major concern is (a) the lack of ablations and feature space visualizations to show the effectiveness of the proposed loss and (b) the contribution over existing works like MAGMA is not enough.
1. The paper is an extension of MAGMA (Merullo et. al. Linearly mapping from image to text space. In ICLR, 2023.). While MAGMA is discussed in the paper, the difference is that this paper with MAGMA is the proposed assignment prediction loss. However, the effectiveness of the proposed loss is not shown clearly in the paper.
2. No ablation results are provided to show the effectiveness of the proposed loss - this is related to weakness-1. Since the major contribution lies in this loss, an ablation to show the contribution of this loss in the final results is very critical.
3. The finding that a linear layer can transform visual representations into language models is not surprising, given existing works LLaVA (“Visual Instruction Tuning”, as in its first training stage), which is not discussed in this paper, and MAGMA as discussed. Therefore, the contribution of this work is weakened.
4. The authors motivate the work by criticizing the “distance-preserving nature of the linear layer”. However, the proposed method is still a linear layer, which doesn’t solve this problem. While Fig-4 provides several examples to show the visual semantic arithmetic, a visualization of feature space would be preferred to show the effects of the assignment loss
5. The paper would be easier to read if the method names (abbreviations) in the results tables come with citations next to them, or are described in texts to show which is which.

**Questions:**

1. Could abalations with and without the assignment loss be provided to show its effectiveness?
2. Could visualizations (e.g. t-SNE) over the feature space with and without the assignment loss be provided, to show its effects in aligning the features?
3. The difference/contribution over LLaVA or MAGMA can be more clearly discussed.

---

> ### Author Response · Authors · 2023-11-20
>
> Thank you for your constructive review and valuable suggestions! Below, we provide a detailed response to your questions and comments. If any of our responses fail to sufficiently address your concerns, please inform us, and we will promptly follow up.
>
> **[W1, W2, Q1] Ablation for the loss function**
>
> We present the ablation analysis corresponding to the learning objectives in the general response.
> Please refer to them.
>
> **[W3, Q3] Difference between VLAP and the previous linear transformation-based approaches**
>
> Thanks for the reference.
> Similar to the image captioning objective of LiMBeR, LLaVa learns the linear projection with the language generation objective, which aims to provide target answers from given transformed visual and language instruction representations.
> Although LiMBeR and LLaVa have shown two unimodal models can be bridged through the language modeling objective, they are insufficient to reduce the modality gap, as shown in our previous response.
> Our proposed objective not only improves overall performance in the linear transformation-based methods, but also significantly facilitates the connection of the visual encoder even if it is not trained with language guidance (i.e., CLIP encoder).
> In addition, our objective can be an alternative to the conventional objectives for cross-modal alignment, such as the image-text matching and image-text contrastive objectives.
> We add the missing reference LLaVa in the main paper.
>
> **[W4, Q2] Linear layer with assignment prediction**
>
> We apologize for using the misleading term, "distance-preserving nature of the linear layer". We revise the Introduction section to clarify the contribution.
>
> **[W5] Model reference**
>
> Thanks for your suggestion. We clarify the model names with citations in the table captions.

---

> ### Author Response · Authors · 2023-11-21
>
> We thank you for your time and effort in reviewing our paper. Your insights have been highly valued, and your feedback is crucial to our progress. We understand that you have a busy schedule, and we again appreciate your time and attention to this matter. We have addressed all of your comments in our rebuttal. We would be grateful if you could please take a look at our rebuttal. If there are any additional materials or information you may require to facilitate the review process, please let us know. Your prompt attention to this matter is greatly appreciated.
>
> Thank you again for your help. Sincerely Authors

---

> ### Author Response · Authors · 2023-11-23
>
> Dear Reviewer zAaX,
>
> Thanks for your constructive and valuable comments on our paper. We have revised our paper to address the reviewer's concerns.
> * **[W1, W2, Q1] Ablation regarding optimal transport.** We added ablation analyses for the components of VLAP, including the hyperparameters and optimal transport in Appendix Section D.
>
> * **[W3, Q3, W4] Difference between VLAP and the previous linear transformation-based approaches.** We included the missing reference (i.e., LLaVA). In addition, we extensively revised the paper to cover the reviewer's concerns about the novelty and to remove potentially misleading terms in Introduction.
>
> * **[W4, Q2] Visualization.** We presented the UMAP visualization corresponding to visual arithmetic and ablation for the learning objectives. We can verify the effectiveness of the proposed VLAP in the feature space.
>
> * **[W5] Citation.** We added references along with the model names to the table captions.
>
> With the discussion deadline approaching in a few hours, we are eager to receive your feedback.

---

### Official Review · Reviewer_dYxb · 2023-10-31

**Soundness:** 2 fair
**Presentation:** 3 good
**Contribution:** 2 fair
**Rating:** 5
**Confidence:** 5

**Summary:**

In vision-language modeling, a significant challenge persists: bridging the modality gap between pretrained vision and language models. This gap arises primarily due to the models' pretraining exclusively on unimodal data, leading to inconsistencies in their embedding spaces. Motivated by this limitation and the computational costs of previous methods, this work introduces VLAP, a novel linear transformation-based approach that employs assignment prediction to connect vision encoders and large language models (LLMs). By harnessing the established word embeddings of LLMs and introducing an optimal transport-based assignment prediction objective, VLAP maps visual data representations to LLM's word embeddings, aiming for consistent modality representation. This not only results in visual data representations with the semantic richness of LLMs but also surpasses prior methods in computational and memory efficiency across various vision-language tasks.

**Strengths:**

1. The limitations from SOA mentioned in the paper exist, and the motivation is valid.
2. Resolving the modality gap problem with the cross-modal assignment prediction using word embeddings of LLMs is a better solution than previous methods.

**Weaknesses:**

1.  A better alignment (reducing the gap) in multi-modality is the essential contribution of this work. However, it lacks studies or results, apart from the overall performance, to validate that the gap reduction is achieved by the current predicted assignments rather than the linear layers from previous works.
2. The authors mentioned, ``Mapping visual data to LLM’s word embeddings results in learned visual representations that hold a semantic taxonomy of LLMs.'' However, there's a lack of quantitative/qualitative results to validate that this allows visual representations to inherit a semantic taxonomy of LLMs.
3. The final objectives are influenced by the assignment prediction loss and captioning loss. However, there's a lack of study on these hyperparameters. Also, which part contributes more to the learning remains a question.
4. For the probability that the corresponding modality data belongs to each word, $P_{nk}$, what does $P_{nk}^{v}$ in the visual modality signify? Does this ``word'' refer to the single word token in the class label of that visual region?
5. There's a lack of formal definitions for the terms/operations appearing in equations, i.e., $Tr(\cdot)$, $[prefix]$.

[Summary] The current limitations and motivations are valid, and the claimed contribution is significant. However, in the paper's delivery, there's a concern about how this performance is achieved by the proposed architecture and mechanism. Additionally, the paper lacks a depth of study beyond introducing a novel architecture.

**Questions:**

Please also refer to the previous section.

---

> ### Author Response · Authors · 2023-11-20
>
> Thank you for your constructive review and valuable suggestions! Below, we provide a detailed response to your questions and comments. If any of our responses fail to sufficiently address your concerns, please inform us, and we will promptly follow up.
>
> **[W1] Validation for the modality gap reduction**
>
> | $\mathcal{L}\_\text{cap}$ | $\mathcal{L}\_\text{ITM}$ | $\mathcal{L}\_\text{ITC}$ | $\mathcal{L}\_\text{map}$ | CIDEr-D | CLIP-S | Ref-S | $\Delta$ |
> |---------------------------|---------------------------|---------------------------|---------------------------|---------|--------|-------|--------------|
> | v                         |                           |                           |                           | 51.3    | 72.4   | 76.6  | 0.972        |
> | v                         | v                         | v                         |                           | 61.9    | 79.7   | 82.8  | 0.870        |
> | v                         | v                         | v                         | v                         | 65.2    | 82.8   | 86.0  | 0.840        |
> | v                         |                           |                           | v                         | 69.4    | 87.6   | 92.0  | 0.725        |
>
> To provide a better understanding in terms of modality gap reduction, we measure the modality gap distance $||\Delta||$ according to the learning objective.
> The modality gap $\Delta$ is defined as the difference between the center of visual and text representations [1]:
>
> $\begin{equation}
>         \Delta = \frac{1}{n}\sum\_{i=1}^{n} \bar{\mathbf{z}}^{v}\_i - \frac{1}{n}\sum\_{i=1}^n \bar{\mathbf{z}}^t\_i,
>     \end{equation}$
>
> where $n$ is the total number of image-text pairs, $\bar{\mathbf{z}}\_i^v$, and $\bar{\mathbf{z}}\_i^t$ are the $i$-th averaged visual and text representations.
> The default gap distance between CLIP ViT-B/32 and T5$\_\text{Base}$ before training is $||\Delta|| = 1.378$.
> The image captioning objective $\mathcal{L}\_\text{cap}$ reduces the modality gap distance to $||\Delta|| = 0.972$ and the additional objectives $\mathcal{L}\_\text{ITM}$ and $\mathcal{L}\_\text{ITC}$ further reduce the gap distance to $||\Delta|| = 0.870$.
> The comparison between VLAP with all objectives and with the original objective (i.e., $\mathcal{L}\_\text{cap}+\mathcal{L}\_\text{map}$) shows the ITM and ITC objectives have a limitation to reduce the modality gap.
> In addition, since the overall performance is inversely proportional to the modality gap, reducing the gap is an important factor in training VLMs.
> We also present UMAP visualization [2] and the additional analysis according to the learning objective in Figure 8 and Appendix. D.2 of the revised paper.
>
>     [1] W. Liang et al., ``Mind the gap: Understanding the modality gap in multi-modal contrastive representation learning," NeurIPS'22
>     [2] T. Sainburg et al., ``Parametric UMAP embeddings for representation and semi-supervised learning," Neural Computation'21

---

> ### Author Response · Authors · 2023-11-20
>
> **[W2] Additional analysis for visual arithmetic**
>
> |         | Building$\rightarrow$Country |       |        | Country$\rightarrow$Capital |      |        | CEO$\rightarrow$Company |       |        | Food$\rightarrow$Country |      |        |
> |---------|------------------------------|-------|--------|-----------------------------|------|--------|-------------------------|-------|--------|--------------------------|------|--------|
> | Method  | B@1                          | R@5   | CLIP-S | B@1                         | R@5  | CLIP-S | B@1                     | R@5   | CLIP-S | B@1                      | R@5  | CLIP-S |
> | ClipCap | 0.003                        | 0.035 | 0.24   | 0.0                         | 0.0  | 0.22   | 0.004                   | 0.005 | 0.18   | 0.0                      | 0.0  | 0.24   |
> | ZeroCap | 0.1                          | 0.32  | 0.7    | 0.14                        | 0.32 | 0.68   | 0.1                     | 0.3   | 0.64   | 0.03                     | 0.33 | 0.66   |
> | VLAP    | 0.3                          | 0.53  | 0.87   | 0.46                        | 0.62 | 0.89   | 0.25                    | 0.5   | 0.75   | 0.17                     | 0.5  | 0.81   |
>
> To prove the visual representation that inherited the semantic taxonomy of LLM, we analyze that word analogies can often be solved with visual vector arithmetic.
> In addition to existing experiments on visual semantic arithmetic, we supplement an experiment on the visual relation (VR) benchmark [3].
> The VR benchmark consists of a total of 320 relations, including building$\rightarrow$country, country$\rightarrow$capital, CEO$\rightarrow$company, food$\rightarrow$country, and leader$\rightarrow$country.
> Since the relation of leader$\rightarrow$country includes out-of-date information (e.g. 'Obama'-'USA'), we exclude it from this experiment.
>     In the above table, we compare the performance between VLAP, ClipCap [4], and ZeroCap [3] in terms of BLEU-1, Recall@5, and CLIP-Score, following [3].
> For fair comparisons, we identically use CLIP ViT-B/32 as the image encoder and GPT-2 as the LLM.
> The results show that VLAP outperforms the previous methods by large margins.
> In particular, we attain correlations of over 75\% in all relations on the semantic distance-based metric, i.e., CLIP-Score.
> We include the additional analysis with visualization in Figure 7 and Appendix. C.2 of the revised paper.
>
>     [3] Y. Tewel et al., ``ZeroCap: Zero-shot image-to-text generation for visual-semantic arithmetic," CVPR'22
>     [4] R. Mokady et al., ``ClipCap: Clip prefix for image captioning," arXiv'21
>
> **[W3] Ablation for the balancing parameters**
>
> We present the ablation results corresponding to the balancing parameters in equation (7) in the general response.
> Please refer to them.
>
> **[W4] Meaning of the probability in equation (5)**
>
> We first explain the probability $\mathbf{P}\_{nk}^{t}$ for the text representation.
> Since LLMs provide text representations through sophisticated operations between word embeddings, text representations and word embeddings have high correlations with each other.
> Therefore, we can obtain the probability of the $k$-th word attending to the $n$-th text representation without any correspondence between language captions and word embeddings as in equation (5).
> Meanwhile, our assignment prediction objective enforces visual and text representations to contain the same information.
> Simply put, the probability $\mathbf{P}\_{nk}^{v}$, which represents the $k$-th word attending to the $n$-th visual representation, is learned to be similar to a pre-calculated probability $\mathbf{P}\_{nk}^t$ by predicting the assignment of text data without any image-word correspondence.
>
> **[W5] Clarity**
>
> We denote the trace matrix as $\text{Tr}(\cdot)$ in equation (2) and the input text prompt (i.e., "$\texttt{A photo of}$") as $\texttt{[prefix]}$ in equation (6).
> We clarify them in the revised paper.

---

> ### Author Response · Authors · 2023-11-21
>
> We thank you for your time and effort in reviewing our paper. Your insights have been highly valued, and your feedback is crucial to our progress. We understand that you have a busy schedule, and we again appreciate your time and attention to this matter. We have addressed all of your comments in our rebuttal. We would be grateful if you could please take a look at our rebuttal. If there are any additional materials or information you may require to facilitate the review process, please let us know. Your prompt attention to this matter is greatly appreciated.
>
> Thank you again for your help. Sincerely Authors

---

> ### Author Response · Authors · 2023-11-23
>
> Dear Reviewer dYxb,
>
> Thanks for your constructive and valuable comments on our paper. We have revised our paper to address the reviewer's concerns.
>
> * **[W1, W3] Ablation regarding objectives.** We added ablation analyses for the learning objectives, including the hyperparameters and additional objectives in Appendix. Section D.
>
> * **[W2] Additional analysis for visual arithmetic.** We added the additional analysis for visual arithmetic on the visual relation benchmark, which demonstrates the learned visual representations can be used to reason semantic relations. We also provided the UMAP visualization to verify how the visual arithmetic of VLAP works in the feature space in Appendix. Section C.
>
> * **[W5] Clarity of notations.** We clarified some notations for readability.
>
> With the discussion deadline approaching in a few hours, we are eager to receive your feedback.

---

### Official Review · Reviewer_5A8P · 2023-10-31

**Soundness:** 3 good
**Presentation:** 4 excellent
**Contribution:** 2 fair
**Rating:** 6
**Confidence:** 5

**Summary:**

This paper introduces VLAP bridges vision encoders and language models through assignment prediction and the use of word embeddings to map visual representations into language space.
An optimal transport-based training objective is proposed to enforce the consistency of word assignments for paired multimodal data. This allows frozen LLMs to ground their word embedding space in visual data and use their robust semantic taxonomy visually.
The experiments demonstrate that VLAP outperforms the linear transformation-based approaches in a variety of vision-language tasks, such as image captioning, visual question answering, and cross-modal retrieval.
It also shows that the visual representations that have been acquired contain a semantic taxonomy of LLMs, thus making it possible to do visual semantic arithmetic.

**Strengths:**

The paper is well-written and easy to follow.
The work proposed a straightforward way of learning the linear projection layer for visual modality to learn multimodal representation, which accommodates the LLM generation.
The visualization shows an impressive semantic arithmetic ability to combine multimodality understanding in LLM generation.

**Weaknesses:**

(1) The main concern of this work is the methodology is relatively incremental without new concepts or findings.
Concept-wise and architecture-wise, it is similar to Asano et al. (2020) Selavi, which performs optimal transport across modalities with similar pipelines. Mathematics using the Sinkhorn clustering Swav as Caron et al. (2020).
(2) The main difference lies in 3 parts: word embedding as fixed center space, different distribution assumptions (polytope), and LLM application.
The first two are the most interesting part, which will be different from previous Sinkhorn-based work.
However, there is no ablation study on these two components, which leads the readers to question whether borrowing existing Selavi and Swav will also work.
(3) Also, there is no ablation on different objectives, such as existing next-word prediction on learning visual projection on LLM.

**Questions:**

Either additional ablation, justification, or additional baseline can elaborate the concern in the weakness (2).

---

> ### Author Response · Authors · 2023-11-20
>
> Thank you for your constructive review and valuable suggestions! Below, we provide a detailed response to your questions and comments. If any of our responses fail to sufficiently address your concerns, please inform us, and we will promptly follow up.
>
> **[W1, W2] Novelty and ablation for assignment prediction**
>
> Thanks for your valuable comments.
> As the reviewer mentioned, our VLAP differs from the previous optimal transport-based (or clustering-based) approaches [1, 2, 3, 4] in the following aspects:
>
> (1) defining a fixed central space instead of a learnable central space (i.e., prototypes) and
> (2) defining the transport polytope with the word distribution of the training dataset instead of the equipartition assumption.
>
> The previous works aim to learn visual representations [1, 2] or multimodal representations [3, 4] by \textbf{training the whole networks}, including backbone networks and the central space, in an end-to-end manner.
> However, we need to connect two \textbf{frozen} backbone networks using visual features as LLMs input.
> In our setup, we empirically found that learnable space is heavily sensitive to the hyperparameters (e.g. $\epsilon$ in equation (4)) and provides poor performance.
> Therefore, the conventional optimal transport-based approaches [1, 2, 3, 4] have been limited to be directly applied to our work.
>
> |       |                      |      | $\epsilon$-MSCOCO (CIDEr-D) |      |       |
> |-------|----------------------|------|-----------------------------|------|-------|
> |       | Method               | 0.1  | 0.05                        | 0.01 | 0.005 |
> | (i)   | VLAP w/o word embed. | 20.4 | 43.8                        | 38.9 | -     |
> | (ii)  | VLAP w/o word dist.  | 50.7 | 57.9                        | 57.1 | 46.2  |
> | (iii) | VLAP                 | 61.8 | 68.0                        | 69.9 | 67.7  |
>
> To verify our claim, we conduct additional experiments for the components in assignment prediction.
> As shown in the above table, we evaluate each model for the zero-shot image captioning performance (CIDEr-D) on MSCOCO corresponding to various $\epsilon$ values in equation (4).
> In (i) "VLAP w/o word embed.'', we first train VLAP with learnable prototypes as in [1, 2, 3, 4].
> In this setting, we define 3K prototypes following [1] and apply the equipartition assumption in equation (3).
> In (ii) "VLAP w/o word dist.'', we also verify the effectiveness of the word distribution.
> In this setting, we use the word embeddings as a fixed central space and perform the optimal transport with the equipartition assumption.
> The comparison between (i) and (ii) shows that the word embedding achieves substantial performance improvement, demonstrating the effectiveness of the word embedding.
> The comparison between (ii) and (iii) original VLAP shows that the transportation polytope with the word distribution achieves additional performance improvement and provides more robust performance regarding $\epsilon$.
>
>     [1] M. Caron et al., ``Unsupervised learning of visual features by contrasting cluster assignments," NeurIPS'20
>     [2] Y. M. Asano et al., ``Self-labeling via simultaneous clustering and representation learning," ICLR'20
>     [3] Y. M. Asano et al., ``Labelling unlabelled videos from scratch with multi-modal self-supervision," NeurIPS'20
>     [4] J. Duan et al., ``Multi-modal alignment using representation codebook," CVPR'22
>
> **[W3] Ablation for the loss function**
>
> Our image captioning objective is basically defined as the next-word generation prediction, i.e., generating a word from the previous information.
> Instead, we provide the results corresponding to the additional objectives for cross-modal alignment in the general responses.
> Please refer to them.

---

> ### Author Response · Authors · 2023-11-21
>
> We thank you for your time and effort in reviewing our paper. Your insights have been highly valued, and your feedback is crucial to our progress. We understand that you have a busy schedule, and we again appreciate your time and attention to this matter. We have addressed all of your comments in our rebuttal. We would be grateful if you could please take a look at our rebuttal. If there are any additional materials or information you may require to facilitate the review process, please let us know. Your prompt attention to this matter is greatly appreciated.
>
> Thank you again for your help. Sincerely Authors

---

> ### Author Response · Authors · 2023-11-23
>
> Dear Reviewer 5A8P,
>
> Thanks for your constructive and valuable comments on our paper. We have revised our paper to address the reviewer's concerns.
> * **[W1, W2] Ablation regarding optimal transport.** We added ablation analyses for the components of VLAP, including the hyperparameters and optimal transport in Appendix Section D.
>
> * **[W3] Ablation regarding learning objectives.** We added ablation analyses with the additional learning objectives in Appendix Section D.
>
> With the discussion deadline approaching in a few hours, we are eager to receive your feedback.

---

> > ### Comment · Reviewer_5A8P · 2023-12-05
> >
> > I have read through the reviews and rebuttals.
> > The author addressed my main concern regarding the novelty and effectiveness of the proposed word embedding optimal transport.
> > The additional analysis for visual arithmetic responding to reviewer dYxb is also interesting. This might be meaningful for other works claiming the model learned visual arithmetic.
> > I'll keep my rating marginally above the acceptance threshold.

---

### Official Review · Reviewer_oDtB · 2023-11-01

**Soundness:** 2 fair
**Presentation:** 3 good
**Contribution:** 3 good
**Rating:** 5
**Confidence:** 4

**Summary:**

The paper proposed to bridge the vision and language modalities by predicting the assignment between LLM word embeddings and those two modalities. Specifically, the optimal transport is employed to decide the assignment between LLM word embeddings and image/caption contextualized embeddings, and then the model is required to predict the assignment of one modality from the other modality. Experiments are conducted on multiple tasks/datasets to prove the effectiveness of the proposed method.

**Strengths:**

1. Demanding one modality's representation to predict the assignment between the other modality and common feature space (LLM word embedding) is an interesting idea to bridge two modalities.
2. Evaluations on different tasks show a better performance than previous work.

**Weaknesses:**

1. Comprehensive ablation of w/ and wo/ assignment prediction on the same vision/language backbones is missing.
2. Comparison with other baselines that are designed for alignment is missing. For example, contrastive alignment in ALBEF, BLIP, and the first-stage alignment by BLIP2 which includes image-text matching, and image-grounded text generation.
3. In experiments, the pre-training data is CC3M which is too small in terms of scale. Whether this method can be generalized to larger scale is not validated.
4. In Tab1,2,3, when compared with previous works, the vision/language backbone is always different. I wonder if using the same backbones as previous works, will the proposed method still outperform them?

**Questions:**

1. Does the LLM word embedding have to be from the same LLM as used in language encoding?

---

> ### Author Response · Authors · 2023-11-20
>
> Thank you for your constructive review and valuable suggestions! Below, we provide a detailed response to your questions and comments. If any of our responses fail to sufficiently address your concerns, please inform us, and we will promptly follow up.
>
> **[W1] Ablation for assignment prediction**
>
> We present the ablation results corresponding to the balancing parameters in equation (7) in the general response. Please refer to that section for more details.
> The result without the assignment prediction objective corresponds to $\lambda\_\text{map}:\lambda\_\text{cap}=0:1$.
> To briefly sum up, the assignment prediction objective combined with $\mathcal{L}\_\text{cap}$ improves the performance.
>
> **[W2] VLAP with other cross-modal alignment**
>
> Thanks for the constructive suggestion.
> We present the results with the additional objectives, including the image-text matching (ITM) and image-text contrastive (ITC) objectives, in the general response.
> Briefly, those two objectives show less performance improvement than our proposed assignment prediction.
>
> **[W3] Scale of the pretraining dataset**
>
> Thank you for your valuable suggestion.
> As previous works for pretraining VLMs have proven, a larger pretraining dataset should bring better results on zero-shot vision-language tasks.
> To verify the impact of the scale of the training dataset on zero-shot performance, we will train VLAP with a large-scale dataset (e.g. CC12M) and will include results in the revised paper.
> However, we would like to emphasize that our method mainly focuses on presenting an efficient methodology (i.e., linear transformation, using pretrained encoders, small-scale training) to make VLMs.
>
> **[W4] VLAP with other backbone**
>
> | Method | Vis. Encoder  | Lang. Model | NoCaps-In | NoCaps-Out | NoCaps-Near | NoCaps-All | NoCaps-CLIP-S | NoCaps-Ref-S | MSCOCO-CIDEr-D | MSCOCO-CLIP-S | MSCOCO-Ref-S |
> |--------|---------------|-------------|-----------|------------|-------------|------------|---------------|--------------|----------------|---------------|--------------|
> | MAGMA  | CLIP RN50x16  | GPT-J       | 30.4      | 43.4       | 36.7        | 38.7       | 74.3          | 78.7         | 47.5           | 75.3          | 79.6         |
> | LiMBeR | CLIP RN50x16  | GPT-J       | 34.3      | 48.4       | 41.6        | 43.9       | 74.7          | 79.4         | 54.9           | 76.2          | 80.4         |
> | VLAP   | CLIP ViT-B/32 | OPT         | 48.2      | 62.7       | 59.3        | 61.3       | 84.8          | 88.5         | 69.9           | 86.7          | 91.8         |
> | VLAP   | CLIP ViT-B/32 | T5          | 48.3      | 62.7       | 59.6        | 61.6       | 85.1          | 88.7         | 69.4           | 87.6          | 92.0         |
> | VLAP   | CLIP RN50x16  | GPT-J       | 53.8      | 67.5       | 65.7        | 64.5       | 88.3          | 90.1         | 75.3           | 90.6          | 72.2         |
>
> While we used the transformer-based vision models (i.e., CLIP ViT-B/32 and BEiT) and the relatively small scale of LLMs (i.e., OPT$\_\text{1.3B}$ and T5$\_\text{Base}$) in the main paper, VLAP can be applied to all publicly available vision models and LLMs.
> Following MAGMA and LiMBeR, we additionally evaluate VLAP with the CNNs-based vision model (i.e., CLIP RN50x16) and the larger LLM (i.e., GPT-J, which has 6B parameters) for zero-shot image captioning on NoCaps and MSCOCO.
> Not surprisingly, VLAP with the larger LLM achieves better performance than with the smaller LLMs, significantly outperforming MAGMA and LiMBeR by large margins across all metrics with the same backbones.
> We add the results to Table. 9 in Appendix.
>
> **[Q1] Consistency of the word embedding in LLMs**
>
> While word embedding of any LLMs can be used, we exploit the word embedding within the LLM used for language encoding for two main reasons:
> (1) LLMs use word embeddings as a basis to produce text representations, indicating text representations already have high correlations with word embeddings.
> Therefore, the consistency of LLM for word embedding and language encoding makes the assignment for language data confident.
> (2) The inconsistency of LLMs for word embedding and language encoding requires unnecessary memory (e.g. OPT$\_\text{1.3B}$ has about 100M parameters for the word embedding).

---

> ### Author Response · Authors · 2023-11-21
>
> We thank you for your time and effort in reviewing our paper. Your insights have been highly valued, and your feedback is crucial to our progress. We understand that you have a busy schedule, and we again appreciate your time and attention to this matter. We have addressed all of your comments in our rebuttal. We would be grateful if you could please take a look at our rebuttal. If there are any additional materials or information you may require to facilitate the review process, please let us know. Your prompt attention to this matter is greatly appreciated.
>
> Thank you again for your help. Sincerely Authors

---

> ### Author Response · Authors · 2023-11-23
>
> Dear Reviewer oDtB,
>
> Thanks for your constructive and valuable comments on our paper. We have revised our paper to address the reviewer's concerns.
> * **[W1, W2] Ablation regarding objectives.** We added ablation analyses for the learning objectives, including the hyperparameters and additional objectives in Appendix Section D.
>
> * **[W4] VLAP with other backbone.** We added the results of VLAP with CLIP RN50x16 and GPT-J, which are the same backbone as the baselinse, MAGMA and LiMBeR in Appendix Section D.
>
> With the discussion deadline approaching in a few hours, we are eager to receive your feedback.

---

> ### Comment · Reviewer_oDtB · 2023-11-23
>
> I have read the author's response and other reviewer's reviews. I appreciate the author's efforts and devotion to solving our concerns. Most of my concerns are solved. However...
>
> I still think it's necessary to experiment with the larger dataset to see how this method performs with the increase in data volume. Many methods, especially for efficient training, don't show performance improvement with larger data scales. These works have merit and contribution in terms of efficient training and fast learning in small/modest data, but it's also important to demonstrate the relationship between performance gain and data scale. Not generalizing to a larger data scale is not a weakness, but we have to let the audience know in which cases/scales our method shines and fails.

---

> > ### Author Response · Authors · 2023-11-23
> >
> > We strongly agree with the reviewer's opinion. Now we are training VLAP on the CC12M and LAION400M datasets to verify the effectiveness of the dataset scale. We apologize for not being able to share the results during this discussion period as the scale of the dataset is extensively large. It is certain that these findings will be included in the final version.

---

### Official Review · Reviewer_m8gW · 2023-11-01

**Soundness:** 3 good
**Presentation:** 3 good
**Contribution:** 3 good
**Rating:** 8
**Confidence:** 3

**Summary:**

The paper aims to align the LLMs (encoder/decoder or just decoder) with image encoders such that the LLMs can comprehend visual input better. It further restricts the design space to freeze the original LLM and visual encoder, just relying on a cheap learned linear transformation. To adapt such a transformation, the paper presents two learning objectives -- assignment prediction and image captioning. Empirical results are presented on 3 different tasks -- image captioning, VQA and cross-modal retrieval (I2T, T2I).

**Strengths:**

* The problem is well motivated with wide applications.

* The paper is mostly well written and explained.

* The empirical results show a big delta which demonstrates the effectiveness of the approach. The studies are also conducted on wide range of problem settings.

**Weaknesses:**

* The motivation for restricting the learned parameter space to just linear layers is unclear -- it would have been more interesting to see more analysis around different learned parameter space including non-linear layers.

**Questions:**

-- Can the authors show ablation studies for the L_map and L_cap objectives to develop better understanding of each component?

---

> ### Author Response · Authors · 2023-11-20
>
> Thank you for your constructive review and valuable suggestions! Below, we provide a detailed response to your questions and comments. If any of our responses fail to sufficiently address your concerns, please inform us, and we will promptly follow up.
>
> **[W1] VLAP with non-linear space**
>
> We would like to emphasize that our method diverges from traditional objectives in linear transformation-based approaches by introducing an alternative objective that avoids direct representation comparisons.
> In addition, although our method dramatically boosted the performance of VLMs based on linear transformation, but nothing is specifically designed for this purpose only.
> As shown in the general response, our assignment prediction objective outperforms other objectives used in the modular-based methods.
> The results suggest that the proposed assignment prediction will contribute to the modular-based methods.
> To address the reviewer's suggestion, we are conducting additional experiments for the non-linear space, such as multiple linear layers with activation functions and an adapter, which is a trainable bottleneck module [1].
> We will include the analysis in the revised paper.
>
>     [1] J. He et al., ``Towards a unified view of parameter-efficient transfer learning," ICLR'22
>
>
> **[Q1] Ablation for the balancing parameters**
>
> Thanks for the question. We present the ablation results corresponding to the balancing parameters in equation (7) in the general response.
> Please refer to them.

---

> ### Author Response · Authors · 2023-11-21
>
> **[W1] VLAP with non-linear space**
>
> |      |                      |                                                   |                |         | MSCOCO |       |
> |------|:--------------------:|:-------------------------------------------------:|:--------------:|:-------:|:------:|:-----:|
> |      |        Method        |                     Objective                     | Train. Params. | CIDEr-D | CLIP-S | Ref-S |
> | (i)  | VLAP w/non-linearity | $\mathcal{L}\_\text{cap}$                         | 1.8M           | 55.6    | 77.1   | 80.5  |
> | (ii) | VLAP w/non-linearity | $\mathcal{L}\_\text{cap}+\mathcal{L}\_\text{map}$ | 1.8M           | 71.1    | 88.9   | 92.4  |
> |  (iii)    | VLAP                 | $\mathcal{L}\_\text{cap}+\mathcal{L}\_\text{map}$ | 0.6M           | 69.4    | 87.6   | 92.0  |
>
> To address the reviewer's suggestion, we conduct additional experiments to evaluate VLAP with the simplest non-linear mapping layers, i.e., three linear layers with the ReLU activation function, as shown in the above table.
> Note that CLIP ViT-B/32 and T5$\_\text{Base}$ are used as the vision model and LLM, respectively.
> The comparison between (i) and (iii) shows that VLAP with the original setting outperforms VLAP with the linear layers, demonstrating that the learning objective has a greater impact on overall performance than the non-linearity.
> Meanwhile, the comparison between (ii) and (iii) shows that the performance can be slightly improved with the non-linear layers when the learning objective is fixed.
> Although we only presented results with the simplest form of non-linear modules, the results suggest that VLAP will show further improved performance when combined with sophisticated modular-based approaches (e.g. Q-Former in BLIP-2).

---

> ### Author Response · Authors · 2023-11-23
>
> **[W1] VLAP with non-linear space (update)**
>
> |      |                      |                                                   |                |         | MSCOCO |       |
> |------|:--------------------:|:-------------------------------------------------:|:--------------:|:-------:|:------:|:-----:|
> |      |        Method        |                     Objective                     | Train. Params. | CIDEr-D | CLIP-S | Ref-S |
> | (i)  | VLAP w/linear layers | $\mathcal{L}\_\text{cap}$                         | 1.8M           | 55.6    | 77.1   | 80.5  |
> | (ii) | VLAP w/linear layers | $\mathcal{L}\_\text{cap}+\mathcal{L}\_\text{map}$ | 1.8M           | 71.1    | 88.9   | 92.4  |
> | (iii) | VLAP w/adapter | $\mathcal{L}\_\text{cap}+\mathcal{L}\_\text{map}$ | 0.4M           | 69.7    | 87.9   | 92.1  |
> |  (iv)    | VLAP                 | $\mathcal{L}\_\text{cap}+\mathcal{L}\_\text{map}$ | 0.6M           | 69.4    | 87.6   | 92.0  |
>
> We update the results of VLAP with the adapter.
> We use a single adapter, which has the dimension of the bottleneck 256 and the ReLU activation between down-/up-projection layers.
> Similar to the comparison between (ii) and (iv), the comparison between (iii) and (iv) shows that the non-linearity improves the performance despite the smaller number of parameters (0.4M vs 0.6M).
> This result consistently suggests that VLAP will show further improved performance when combined with sophisticated modular-based approaches (e.g. Q-Former in BLIP-2).

---

### Author Response · Authors · 2023-11-20
**General response to the reviewers**

We thank the reviewers for their positive evaluation and constructive comments.
    We are pleased that the reviewers recognized that VLAP is $\textbf{well-motivated}$ (m8gW, dYxB), the problem is $\textbf{clearly stated}$ (m8gW, 5A8P, zAaX), and the idea that uses LLMs' word embeddings is $\textbf{interesting}$ (oDtB).
    We notice that the main concerns of the reviewers come from the lack of ablation studies.
    During the rebuttal period, we conducted extensive ablation studies and enhanced the overall completeness through meticulous paper proofreading, notably in the Introduction section. These efforts aimed to enhance readability and clearly articulate the novelty of our work.
    We would like to first address the concerns about the learning objective that all reviewers commonly pointed out.

**Ablation for the balancing parameters**

| $\lambda_\text{map}$ | $\lambda_\text{cap}$ | CIDEr-D | CLIP-S | Ref-S |
|-----------------------|----------------------|---------|--------|-------|
| 0                     | 1                    | 51.7    | 72.1   | 76.3  |
| 0.2                   | 0.8                  | 62.3    | 80.3   | 85.0  |
| 0.4                   | 0.6                  | 68.1    | 84.8   | 89.2  |
| 0.5                   | 0.5                  | 69.6    | 86.3   | 91.6  |
| 0.6                   | 0.4                  | 69.9    | 86.7   | 91.8  |
| 0.8                   | 0.2                  | 64.5    | 83.1   | 87.8  |
| 1                     | 0                    | 49.6    | 72.4   | 76.1  |

We report the zero-shot image captioning performance on MSCOCO corresponding to different ratios between $\lambda\_\text{map}$ and $\lambda\_\text{cap}$ in equation (7) while keeping the sum of two terms as 1.
For example, the final objective function of the first row (i.e., $\lambda\_\text{map}:\lambda\_\text{cap} = 0:1$) is $\mathcal{L} = \mathcal{L}\_\text{cap}$, which is equivalent to the learning objective of LiMBeR [1] and the first stage of LLaVA [2].
With the language modeling objective, VLAP shows significantly poor performance, attaining 51.7 of CIDEr-D.
Increasing the ratio of $\lambda_\text{map}$ gradually improves the performance, achieving the best performance of 69.9 with $\lambda\_\text{map}:\lambda\_\text{cap} = 0.6:0.4$.
Meanwhile, VLAP with only assignment prediction (i.e., $\mathcal{L} = \mathcal{L}\_\text{map}$) performs worse than with only language modeling, showing a performance of 49.6.
The results demonstrate the effectiveness of the proposed assignment prediction objective while emphasizing the necessity of the language modeling objective to leverage the generative ability of LLMs.
We add those results in Table. 7(a) and discussion in Appendix. D.

    [1] J. Merullo et al., ``Linear mapping from image to text space," ICLR'23
    [2] H. Liu et al., ``Visual instruction tuning," NeurIPS'23

---

> ### Author Response · Authors · 2023-11-20
> **General response to the reviewers**
>
> **VLAP with other cross-modal alignment objectives**
> | $\mathcal{L}\_\text{cap}$ | $\mathcal{L}\_\text{ITM}$ | $\mathcal{L}\_\text{ITC}$ | $\mathcal{L}\_\text{map}$ | CIDEr-D | CLIP-S | Ref-S |
> |--------------------------|------------------------|------------------------|------------------------|---------|--------|-------|
> | v                   |                        |                        |                        | 51.7    | 72.1   | 76.3  |
> | v                   | v                 |                        |                        | 58.4    | 78.2   | 81.7  |
> | v                   | v                 | v                 |                        | 61.8    | 79.0   | 82.3  |
> | v                   | v                 | v                 | v                 | 65.7    | 82.5   | 85.8  |
> | v                   |                        |                        | v                 | 69.9    | 86.7   | 91.8  |
>
> We provide the ablation analysis with the image-text matching (ITM) and image-text contrastive (ITC) objectives used in [3, 4, 5].
>     We modify the ITC and ITM objectives to evaluate the performance of VLAP trained with these objectives.
>     For ITC, we contrast the similarity between averaged image and text representations of a positive pair against those of negative pairs in a batch.
>     The ITC objective $\mathcal{L}\_\text{ITC}$ is defined as the cross-entropy loss between pairwise similarities and the groundtruth one-hot similarities, where the positive pair has 1 and the negative pairs have 0.
>     For ITM, we learn an additional linear layer as a binary classifier to determine whether an image-text pair is matched or not.
>     We concatenate the averaged image and text representations and feed them into the classifier with the softmax activation.
>     Following [3, 4, 5], we also employ the hard negative sampling strategy based on the pairwise similarity.
>     As shown in the above table, the ITM and ITC objectives contribute to performance improvement with $\mathcal{L}\_\text{cap}$.
>     However, we notice that the ITM and ITC objectives hurt overall performance, as shown in the performance comparison between the last two rows in the table.
>     We carefully argue that ITM and ITC solely focus on direct pairwise comparisons without considering the semantic correlation between predefined two embedding spaces.
>     Namely, they enforce instance discrimination and encourage the existence of the modality gap.
>     By comparing the cross-modal intermediate assignments, we can relax the instance discrimination task, effectively reducing the modality gap.
>     We add those results in Table. 7(b) and discussion in Appendix. D.
>
>     [3] J. Li et al., ``BLIP-2: Bootstrapping language-image pre-training with frozen image encoders and large language models," ICML'23
>     [4] J. Li et al., ``Align before fuse: Vision and language representation learning with momentum distillation," NeurIPS'21
>     [5] J. Li et al., ``BLIP: Bootstrapping language-image pre-training for unified vision-language understanding and generation," arXiv'22
>
> Next, we carefully address each reviewer's remaining concerns.

---

### Meta-Review · Area_Chair_H155 · 2023-12-04

**Metareview:**

The paper proposes VLAP, a method to bridge vision encoders and language models through assignment prediction. VLAP achieves substantial improvements in various vision-language tasks and demonstrates good performance in visual semantic arithmetic.

Overall, Reviewers m8gW and 5A8P provided positive reviews, finding the proposed method interesting and acknowledging its achievement of state-of-the-art results. Reviewer 5A8P highlighted the method's incremental nature compared to existing work and requested ablation tests to justify the impact of these changes. The authors addressed these comments during the rebuttal process, which I found satisfactory. However, neither of these reviewers responded to the authors' rebuttal.

Reviewers dYxb, zAaX, and oDtB offered slightly negative reviews. Their main concern, among others, was the proposed method is incremental, despite its strong results across various tasks. The authors responded comprehensively to all reviewer comments, which I found satisfactory. Only Reviewer oDtB responded to the authors' rebuttal, expressing satisfaction with the response and requesting experiments on larger datasets. The authors acknowledged this need but, despite their intent to conduct these experiments, it did not change Reviewer oDtB's rating.

Considering the above discussion and the rebuttal/changes made to the paper, I recommend acceptance. Although the idea might be viewed as incremental, the experiments showcase strong results that could significantly impact the machine learning community specializing in this topic.

**Justification For Why Not Higher Score:**

The rebuttal didn't considerably alter the reviewers' rating.

**Justification For Why Not Lower Score:**

A lower score would typically result in the rejection of the paper; however, this paper holds the potential to make an impact on the ML community focusing on the topic.

---

### Decision · Program_Chairs · 2024-01-16

Accept (poster)